# When Rigid Coherency Hurts: Distributional Coherency Regularization for Probabilistic Hierarchical Time Series Forecasting

## Abstract

Probabilistic hierarchical time-series forecasting is an important variant of time-series forecasting, where the goal is to model and forecast multivariate time-series that have hierarchical relations. Previous works all assume rigid consistency over the given hierarchies and do not adapt to real-world data that show deviation from this assumption. Moreover, recent state-of-art neural probabilistic methods also impose hierarchical relations on point predictions and samples of predictive distribution. This does not account for full forecast distributions being coherent with the hierarchy and leads to poorly calibrated forecasts. We close both these gaps and propose PROFHIT, a probabilistic hierarchical forecasting model that jointly models forecast distributions over the entire hierarchy. PROFHIT (1) uses a flexible probabilistic Bayesian approach and (2) introduces *soft distributional coherency regularization* that enables end-to-end learning of the entire forecast distribution leveraging information from the underlying hierarchy. This enables robust and calibrated forecasts as well as adaptation to real-life data with varied hierarchical consistency. PROFHIT provides 41-88% better performance in accuracy and 23-33% better calibration over a wide range of dataset consistency. Furthermore, PROFHIT can robustly provide reliable forecasts even if up to 10% of input time-series data is missing, whereas other methods' performance severely degrade by over 70%.

## 1 Introduction

Time-series forecasting is an important problem that impacts decision-making in a wide range of applications. In many real-world situations, the time-series have inherent hierarchical relations and structures. Examples include forecasting time-series of employment (Taieb et al., 2017) measured at different geographical scales; epidemic forecasting (Reich et al., 2019) at county, state and country, etc. Given time-series dataset with underlying hierarchical relations, the goal of Hierarchical Time-series Forecasting (`HTSF`) is to generate accurate forecast for all time-series leveraging the hierarchical relations between time-series (Hyndman et al., 2011).

Most previous methods do not provide *well-calibrated* forecasts for *both* so-called *"strong"* and *"weakly" consistent* datasets. Previous `HTSF` methods assume that the time-series values of datasets strictly satisfy the underlying hierarchical constraints and impose *rigid coherency* on generated forecasts i.e., forecasts strictly satisfy the hierarchical relations of dataset. These methods can model datasets generated (Taieb et al., 2017) by first collecting data for time-series of the leaf level nodes and deriving time-series for higher-level nodes. We call such data as *strongly consistent*. For example, classical `HTSF` methods (Hyndman & Athanasopoulos, 2018) use a bottom-up or top-down approach where all time-series at a single level of hierarchy are modeled independently and the values of other levels are derived using the aggregation function governing the hierarchy. In contrast, many real-world datasets are *weakly consistent*, i.e., they do not follow the strict constraints of the hierarchy[1]. Such data have an underlying data generation process that may follow a hierarchical set of constraints but may contain some deviations. These deviations can be caused by factors such as measurement or reporting error, asynchrony in data aggregation and revision pipeline, etc, as frequently observed in

---

[1]Note that we describe *consistency* over a dataset and *coherency* over model forecasts.

epidemic forecasting (Adhikari et al., 2019). Most state-of-the-art `HTSF` methods are designed for strongly consistent datasets and impose rigid coherency constraints — they thus may not adapt to such deviations and can *provide poor forecasts for weakly consistent datasets*.

Moreover, previous methods do not focus on providing *calibrated forecasts* with precise uncertainty measures. Traditional methods focus on point predictions only. Recent methods like MINT (Wickramasuriya et al., 2019), ERM (Ben Taieb & Koo, 2019) and PEMBU (Taieb et al., 2017) refine raw independent forecast distribution as a post-processing step. This does not enable the models generating the raw forecasts to leverage underlying hierarchical relations across time-series. End-to-end learning neural methods directly leverage hierarchical relations as part of the model architecture or learning algorithm like HIERE2E (Rangapuram et al., 2021) and SHARQ

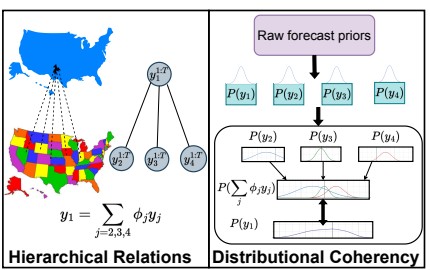

Figure 1: Regularizing forecasts using Distributional Coherency.

(Han et al., 2021). They do this and usually outperform post-processing methods by imposing hierarchical constraints on the mean or fixed quantiles of the forecast distributions. However, these methods do not enforce hierarchical coherency on the full distributions. Therefore, the *forecasts may not be well-calibrated* (Kuleshov et al., 2018) i.e., they produce unreliable prediction intervals that may not match observed probabilities from ground truth (Fisch et al., 2022).

In this work, we fill this gap of learning well-calibrated and accurate forecasts for both strong and weakly consistent datasets leveraging underlying hierarchical relations. We propose PROFHIT (Probabilistic Robust Forecasting for Hierarchical Time-series), a neural probabilistic `HTSF`

Table 1: Comparison of PROFHIT with state-of-the-art methods.

|  | 2-step methods | MINT erm | PEMBU | HIERE2E | SHARQ | **PROFHIT** (This paper) |
|---|---|---|---|---|---|---|
| Probabilistic Forecasts | × | ✓ | ✓ | ✓ | ✓ | ✓ |
| Strong & Weak Consistency | × | × | × | × | ✓ | ✓ |
| Distributional Coherency | × | × | ✓ | × | × | ✓ |
| End-to-end Learning | × | × | × | ✓ | ✓ | ✓ |
| Robust to missing data | × | × | × | × | × | ✓ |

method that provides an end-to-end Bayesian approach to model the distributions of forecasts of all time-series together (see Table 1 for a comparison). Specifically, we introduce a novel *Soft Distributional Coherency Regularization (SDCR)* to tackle the challenge. First, SDCR enables PROFHIT to leverage hierarchical relations over entire forecast distributions to generate calibrated forecast distributions by encouraging forecast distribution of any parent node to be similar to aggregation of children nodes' forecast distribution (Figure 1). Second, since SDCR is a soft constraint, our model is trained to adapt to datasets with varying hierarchical consistency that allows the model to trade-off coherency for better accuracy and calibration on weakly consistent datasets. Our main contributions are:

**(1) Accurate and Calibrated Probabilistic Hierarchical Time-Series Forecasting:** We propose PROFHIT, a deep probabilistic framework for modeling the distributions of each time-series together using a soft distributional coherency regularization (SDCR). PROFHIT leverages probabilistic deep-learning models to learn priors of individual time-series and refines the priors of all time-series leveraging the hierarchy to provide accurate and well-calibrated forecasts.

**(2) Adaptation to Strong and Weak Consistency via Soft Distributional Coherency Regularization:** SDCR imposes soft hierarchical constraints on the full forecast distributions to help adapt the model to varying levels of hierarchical consistency. We build a novel refinement module over raw forecast priors and leverage multi-task learning over shared parameters that enable PROFHIT to perform consistently well across the hierarchy.

**(3) Evaluation Across Multiple Datasets and with Missing Data:** We show that our method PROFHIT outperforms a wide variety of state-of-the-art baselines on both accuracy and calibration, at all levels of the hierarchy, for both strong and weakly consistent datasets. We also show training using SDCR enables PROFHIT to leverage hierarchical relations to provide robust predictions that can handle missing data values in the time-series.

## 2 PROBLEM STATEMENT

Consider the dataset $\mathcal{D}$ of $N$ time-series over the time horizon $1, 2, \ldots, T$. Let $\mathbf{y}_i \in \mathbb{R}^T$ be time-series $i$ and $y_i^{(t)}$ its value at time $t$. The time-series have a hierarchical relationship denoted as $\mathcal{T} = (G_{\mathcal{T}}, H_{\mathcal{T}})$ where $G_{\mathcal{T}}$ is a tree of $N$ nodes rooted at time-series 1. For a non-leaf node (time-series) $i$, we denote its children as $\mathcal{C}_i$. The node values are related via set of relations $H_{\mathcal{T}}$ of form $H_{\mathcal{T}} = \{\mathbf{y}_i = \sum_{j \in \mathcal{C}_i} \phi_{ij} \mathbf{y}_j : \forall i \in \{1, 2, \ldots, N\}, |\mathcal{C}_i| > 0\}$ where values of $\phi_{ij}$ are known and time-independent real-valued constants.

**Definition 1** (Consistency Error - CE). *Given a dataset $\mathcal{D}$ of $N$ time-series over the time horizon $1, 2, \ldots, T$ and aggregation relations $H_{\mathcal{T}}$ as above, the dataset consistency error (CE) is defined as*

$$E_{\mathcal{T}}(\mathcal{D}) = \sum_{i \in \{1, 2, \ldots N\}, \mathcal{C}_i \neq \emptyset} \left( \mathbf{y}_i - \sum_{j \in \mathcal{C}_i} \phi_{ij} \mathbf{y}_j \right)^2. \tag{1}$$

*(Intuitively, datasets with lower CE have time-series values which more strictly follow relations $H_{\mathcal{T}}$).*

**Definition 2** (Strong and weak consistency). *A dataset $\mathcal{D}$ is strongly consistent if $E_{\mathcal{T}}(\mathcal{D}) = 0$. Otherwise, $\mathcal{D}$ is said to be weakly consistent.*

Let current time-step be $t$. For any $1 \leq t_1 < t_2 \leq t$, we denote $\mathbf{y}_i^{(t_1:t_2)} = \{y_i^{(t_1)}, y_i^{(t_1+1)}, \ldots, y_i^{(t_2)}\}$. Given the data $\mathcal{D}^t = [\mathbf{y}_1^{1:t}, \mathbf{y}_2^{1:t}, \ldots, \mathbf{y}_N^{1:t}]$ and hierarchical relations $H_{\mathcal{T}}$, a model $M$ is trained to predict the marginal forecast distributions at time $t + \tau$ for all time-series of hierarchy leveraging past values of all time-series: $\{p_M(y_1^{(t+\tau)}|\mathcal{D}^t), \ldots p_M(y_N^{(t+\tau)}|\mathcal{D}^t)\}$. Along with accuracy of probabilistic forecasts we also evaluate forecast distributions for *calibration*. We define calibration of model forecasts based on previous works (Kamarthi et al., 2021; Kuleshov et al., 2018):

**Definition 3.** *(Calibration Score of Model) Given a model $M$ we define a calibration function $k_M : [0, 1] \to [0, 1]$ as follows: Given a confidence $c$, $k_M(c)$ is the fraction of the predictions for which the ground truth lies within $c$-confidence interval. The calibration score $CS(M)$ is the total deviation between $c$ and $k_M(c)$: $CS(M) = \int_0^1 |k_M(c) - c| dc$. A perfectly calibrated model is such that $\forall c : k_M(c) \approx c$.*

Given a dataset $\mathcal{D}$ with underlying hierarchical relations $H_{\mathcal{T}}$, the goal of *Calibrated Probabilistic Hierarchical Forecasting* is to design a model $M$ that provides accurate and well-calibrated forecast distributions $\{p_M(y_1^{(t+\tau)}|\mathcal{D}^t), \ldots p_M(y_N^{(t+\tau)}|\mathcal{D}^t)\}$ across all levels of the hierarchy for both weakly and strongly consistent datasets.

## 3 METHODOLOGY

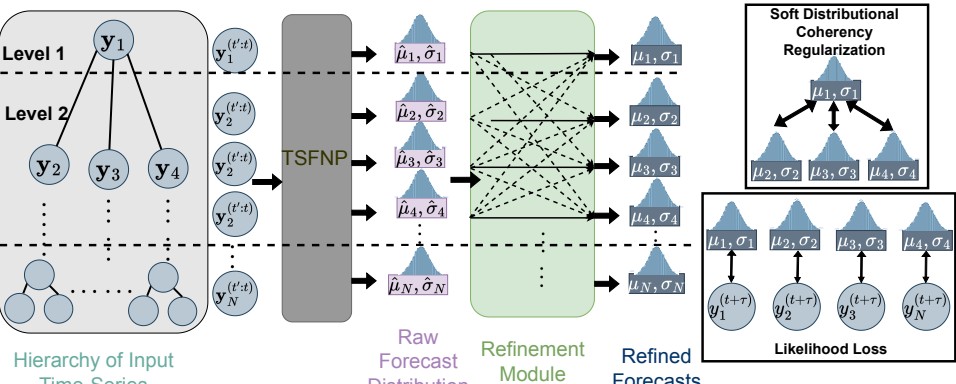

Figure 2: Overview of pipeline of PROFHIT. The input time series is ingested by TSFNP, a Neural Gaussian Process based probabilistic forecasting model to output the raw forecast distribution. The parameters of raw forecasts are refined by the Refinement module using predictions from all time-series. The training is driven by a likelihood loss that learns from ground truth and Soft Distributional Coherency Regularization that regularizes the forecast distribution to follow the hierarchical relations.

**Overview**   PROFHiT models the forecast distributions $\{P(y_i^{(t+\tau)}|\mathcal{D}^t)\}_{i=1}^N$ of all time-series nodes of the hierarchy by leveraging the relations from the hierarchy to provide accurate and well-calibrated forecasts that are adaptable to varying hierarchical consistency. Most existing methods do not attempt to model entire probabilistic distribution but focus on coherency of point forecasts or samples or fixed quantiles of the distribution (Rangapuram et al., 2021; Han et al., 2021). This approach does not fully capture the uncertainty of the forecasts and in turn does not provide calibrated predictions. Moreover, most methods operate on datasets that are strongly consistent over hierarchical relations. However, many real-world datasets are weakly consistent with time-series values of all nodes of hierarchy observed simultaneously and may not follow the hierarchical relations strictly due to noise and discrepancies in collecting data at different levels. Therefore, most previous works may not adapt well to such deviations from these constraints.

PROFHiT, on the other hand, reconciles the need to model coherency between entire forecast distributions as well as induce a soft adaptable constraint to enforce coherency via a two-step stochastic process that is trained in an end-to-end manner. PROFHiT first produces a *raw forecast* distribution for each node parameterized by $\{(\hat{\mu}_i, \hat{\sigma}_i)\}_{i=1}^N$ by using past values of time-series via a neural probabilistic forecasting model. Raw forecasts of all nodes are used as priors to derive a refined set of forecast distributions parameterized by $\{(\mu_i, \sigma_i)\}_{i=1}^N$ via the *refinement module*. The full probabilistic process of PROFHiT is depicted in Figure 2 and formally summarized as:

$$
P(\{y_i^{(t+\tau)}\}_{i=1}^N|\mathcal{D}^t) = \int \underbrace{P(\mathbf{z}|\{\mathbf{y}_i^{(1:t)}\}_{i=1}^N)\left(\prod_{i=1}^N P(\mathbf{z}_i, \mathbf{u}_i|\{\mathbf{y}_i^{(1:t)}\}_{i=1}^N)P(\hat{\mu}_i, \hat{\sigma}_i|\mathbf{z}_i, \mathbf{z}, \mathbf{u}_i)\right)}_{\text{TSFNP (Raw forecasts)}}
$$

$$
\underbrace{\prod_{i=1}^N P(\mu_i, \sigma_i|\{\hat{\mu}_j, \hat{\sigma}_j\}_{j=1}^N)P(y_i^{(t+\tau)}|\mu_i, \sigma_i)}_{\text{Refinement Module}} \, d\{\mathbf{u}_i\}_{i=1}^N d\{\mathbf{z}_i\}_{i=1}^N. \tag{2}
$$

where $\mathbf{z}_i, \mathbf{u}_i, \mathbf{z}$ are intermediate latent variables of our probabilistic raw forecasting model TSFNP (Section 3.1). PROFHiT's SDCR regularizes the parameters $\{(\mu_i, \sigma_i)\}_{i=1}^N$ to leverage the hierarchical relations by minimizing the Distributional Coherency Error (DCE) defined as follows:

**Definition 4.** *(Distributional Coherency Error - DCE) Given the forecasts at time $t + \tau$ as* $\{p_M(y_1^{(t+\tau)}|\mathcal{D}^t), \ldots p_M(y_N^{(t+\tau)}|\mathcal{D}^t)\}$ *distributional coherency error (DCE) is defined as*

$$
\sum_{i \in \{1,\ldots,N\}, \mathcal{C}_i \neq \emptyset} Dist\left(p_M(y_i^{(t+\tau)}|\mathcal{D}^t), p_M(\sum_{j \in \mathcal{C}_i} \phi_{i,j} y_j^{(t+\tau)}|\mathcal{D}^t)\right) \tag{3}
$$

*where $Dist$ is a distributional distance metric.*

Leveraging distributional coherency error as a soft regularizer enforces forecast distributions to be well-calibrated while adaptively adhering to hierarchical relations of the dataset.

## 3.1   RAW FORECAST DISTRIBUTIONS FROM NGPS

NGPs (Neural Gaussian Process) (Louizos et al., 2019) are novel class of probabilistic neural models state-of-the-art accurate and calibrated predictions. We, therefore, use a modified form of a state-of-the-art NGP model for time-series forecasting (Kamarthi et al., 2021) which we call TSFNP to derive raw forecast distributions for each time-series.

We briefly describe TSFNP's three components: 1) *Probabilistic Neural Encoder*: It encodes the input univariate time-series into a latent stochastic embedding via a GRU (Cho et al., 2014) followed by a self-attention layer (Vaswani et al., 2017):

$$
[\mu(\mathbf{u})_i, \log \sigma(\mathbf{u})_i] = \text{Self-Atten}(\text{GRU}(\mathbf{y}_i^{(t':t)})), \quad \mathbf{u}_i \sim \mathcal{N}(\mu(\mathbf{u})_i, \sigma(\mathbf{u})_i). \tag{4}
$$

2) *Stochastic Data Correlation Graph*: We further leverage similar patterns of past time-series data and aggregate them as **local latent variable**. Unlike EPiFNP which uses past time-series information from same node, in our multi-variate case TSFNP uses past information from all nodes. Formally, for input sequence $\mathbf{y}_i^{(t':t)}$ and each of the past sequence $\mathbf{y}_j$ where $j \in \{1, \ldots, N\}$, we sample $\mathbf{y_j}$ with probability $\exp(-\gamma||\mathbf{u}_i - \mathbf{u}_j||_2^2)$ into set $N_i$. Then, we derive the local latent variable as

$$
\mathbf{z}_i \sim \mathcal{N}\left(\sum_{j \in N_i} \Theta_1(\mathbf{u}_j), \exp(\sum_{j \in N_i} \Theta_2(\mathbf{u}_j))\right) \tag{5}
$$

where $\Theta_1$ and $\Theta_2$ are feed-forward networks.

3) *Predictive Distribution Decoder*: Finally, we combine the latent embedding of input time-series, local latent variable and combined information of all past sequences to derive the parameters of the output distribution via a simple feed-forward network. We first derive a *global latent variable* that combines the information from latent embeddings of all past sequences via self-attention:

$$\{\beta_i\}_{i=1}^N = \text{Self-Atten}(\{\mathbf{u}_i\}_{i=1}^N), \quad \mathbf{z} = \sum_{i=1}^N \beta_i \mathbf{u}_i \tag{6}$$

Finally, we combine the latent embedding of input time-series, local latent variable and global latent variable to derive the raw forecast distribution modelled as a Gaussian $\mathcal{N}(\hat{\mu}_i, \hat{\sigma}_i)$ as:

$$\mathbf{e} = \text{concat}(\mathbf{u}_i, \mathbf{z}_i, \mathbf{z}), \quad [\hat{\mu}_i, \log \hat{\sigma}_i] = \Theta_3(\mathbf{e}) \tag{7}$$

where $\Theta_3$ is a feed forward network.

## 3.2 REFINEMENT MODULE

The refinement module leverages the raw distributions of all nodes of hierarchy to produce refined forecast distributions using hierarchical relations. Given the parameters of *raw forecast* distributions $\{\hat{\mu}_i, \hat{\sigma}_i\}_{i=1}^N$ derived from TSFNP for all time-series $\{\mathbf{y}_i^{(t':t)}\}_{i=1}^N$, the refinement module derives the refined forecast distributions denoted by parameters $\{\mu_i, \sigma_i\}_{i=1}^N$ as functions of parameters of raw forecasts of all time-series. The refined forecasts are optimized to be more coherent using SDCR. Since we drive the full distributions of refined forecasts to be coherent, rather than just the samples or mean statistics, the refined distributions' calibration is also consistent with the hierarchical relations.

Let $\hat{\mu} = [\hat{\mu}_1 \ldots, \hat{\mu}_N]$ and $\hat{\sigma} = [\hat{\sigma}_1 \ldots, \hat{\sigma}_N]$ be vectors of means and standard deviations of raw distributions. We model the refined mean as a function of the raw means of all the nodes. Formally, we derive the mean $\mu_i$ of refined distribution as a weighted sum of two terms: a) $\hat{\mu}_i$, the mean of raw time-series, and b) linear combination of all raw mean of all time-series:

$$\gamma_i = \text{sigmoid}(\hat{w}_i), \quad \mu_i = \gamma_i \hat{\mu}_i + (1 - \gamma_i) \mathbf{w}_i^T \hat{\mu}. \tag{8}$$

$\{\hat{w}_i\}_{i=1}^N$ and $\{\mathbf{w}_i\}_{i=1:N}$ are both learnable set of parameters of the model. $\text{sigmoid}(\cdot)$ denotes the sigmoid function. $\gamma_i$ helps model the trade-off between the influence of the raw distribution of node $i$ and the influence of the other nodes of the hierarchy. This is useful for the model to automatically adapt to datasets with varying hierarchical consistency.

Similarly, we assume the variance of the refined distribution depends on the raw mean and variance of all the time-series. The variance parameter $\sigma_i$ of the refined distribution is derived from the raw distribution parameters $\hat{\mu}$ and $\hat{\sigma}$ as

$$\sigma_i = c\hat{\sigma}_i \text{sigmoid}(\mathbf{v}_{1i}^T \hat{\mu} + \mathbf{v}_{2i}^T \hat{\sigma} + b_i) \tag{9}$$

where $\{\mathbf{v}_{1i}\}_{i=1}^N$, $\{\mathbf{v}_{2i}\}_{i=1}^N$ and $\{b_i\}_{i=1}^N$ are parameters and $c$ is a positive constant hyperparameter.

## 3.3 LIKELIHOOD LOSS AND REGULARIZATION OVER HIERARCHY

We optimize the probabilistic process of Equation 2 for accuracy and calibration by leveraging hierarchical relations by training on likelihood loss on ground truth training data as well as SDCR.

**Likelihood Loss** To maximise the likelihood of forecasts over ground truth $P(\{y_i^{(t+\tau)}\}_{i=1}^N | \mathcal{D}_t)$ we use variational inference by approximating the posterior $\prod_{i=1}^N P(\mathbf{z}_i, \mathbf{u}_i^{(j)} | \mathcal{D}_t)$ with the variational distribution $\prod_{i=1}^N P(\mathbf{u}_i | \mathbf{y}_i^{(t':t)}) q_i(\mathbf{u}_i | \mathbf{y}_i^{(t':t)})$ where $q_i$ is a feed-forward network over GRU hidden embeddings of Probabilistic Neural Encoder that parameterizes the Gaussian distribution of $q_i(\mathbf{u}_i | \mathbf{y}_i^{(t':t)})$. We derive the ELBO (detailed derivation in Appendix) as

$$\mathcal{L}_1 = -E_{\prod_i q_i(\mathbf{z}_i, \mathbf{u}_i | \mathcal{D}_t)}[\log P(\{y_i^{(t+\tau)}\}_{i=1}^N | \{\mathbf{u}_i, \mathbf{z}_i\}_{i=1}^N, \mathbf{z}) + \sum_{i=1}^N \log P(\mathbf{z}_i | \mathbf{u}_i, \{\mathbf{u}_j\}_{j=1}^N) - \log q_i(\mathbf{u}_i | \mathbf{y}_i^{(t':t)})].$$

$$\tag{10}$$

**Soft Distributional Coherency Regularization** PROFHIT leverages the hierarchy relations in $\mathcal{T}$ and regularizes the refined distributions to be coherent. Since PROFHIT aims to leverage hierarchical coherency for improved robustness and calibration, we regularize over the full distributions by using *distributional coherency error* as part of the loss function. We use the Jensen-Shannon Divergence (Endres & Schindelin, 2003) (JSD) as the distance metric since it is a symmetric and bounded

variant of the popularly used KL-Divergence distance and assumes closed form for many widely used distributions. We derive the *distributional coherency error* on $\{(\mu_i, \sigma_i)\}_{i=1}^N$ as

$$\mathcal{L}_2 = 2 \left[ \sum_{i=1}^N \text{JSD} \left( P(y_i^{(t+\tau)} | \mu_i, \sigma_i), P \left( \sum_{j \in \mathbf{C}_i} \phi_{ij} y_j^{(t+\tau)} | \{\mu_j, \sigma_j\}_{j \in \mathbf{C}_i} \right) \right) + 1 \right]. \tag{11}$$

Computation of JSD is generally intractable. However, in our case, due to parameterization of each time-series distribution as a Gaussian we get a closed-form differentiable expression:

$$\mathcal{L}_2 = \sum_{i=1}^N \frac{\sigma_i^2 + \left( \mu_i - \sum_{j \in C_i} \phi_{ij} \mu_j \right)^2}{2 \sum_{j \in C_i} \phi_{ij}^2 \sigma_j^2} + \sum_{i=1}^N \frac{\sum_{j \in C_i} \phi_{ij}^2 \sigma_j^2 + \left( \mu_i - \sum_{j \in C_i} \phi_{ij} \mu_j \right)^2}{2 \sigma_i^2}. \tag{12}$$

We provide the derivation of Equation 12 in the Appendix. We use the distributional coherency error as a soft regularization term to enable PROFHIT to leverage constraints $H_{\mathcal{T}}$ when generating forecast distributions. Thus, the total loss for training is given as $\mathcal{L} = \mathcal{L}_1 + \lambda \mathcal{L}_2$ where the hyperparameter $\lambda$ controls the trade off between data likelihood and coherency. We also use the reparameterization trick to make the sampling process differentiable and we learn the parameters of all training modules via Stochastic Variational Bayes (Kingma & Welling, 2013). The full pipeline of PROFHIT is summarized in Figure 2.

### 3.4 DETAILS ON TRAINING

**Parameter sharing across nodes** Since PROFHIT's TSFNP module forecasts for multiple nodes, we leverage the hard-parameter sharing paradigm of multi-task learning (Caruana, 1997) and use different set of parameters for Predictive Distribution Decoder (i.e., weights of $\Theta_3$ for each time-series $i$ is different) whereas the parameters of other components of TSFNP are shared across all nodes (Figure 2). Sharing parameters for Probabilistic Neural Encoder drastically lowers the number of learnable parameters since datasets can have large number of nodes (up to 512 nodes in our experiments).

**Pre-training on individual time-series** Before we start training for refined forecasts, we pre-train the parameters of TSFNP on given training dataset to model raw forecast distribution accurately. We train using only a log likelihood loss to learn parameters $\{\hat{\mu}_i, \hat{\sigma}_i\}_{i=1}^N$ similar to Equation 10.

## 4 EXPERIMENTS

We evaluate PROFHIT over multiple datasets and compare it with state-of-the-art baselines[2].

### 4.1 SETUP

**Baselines:** We compare PROFHIT's performance against state-of-the-art HTSF methods. We also compare against state-of-the-art general probabilistic forecasting methods to study the importance of modeling the hierarchy for both weak and strongly consistent datasets. (1) **TSFNP** (Kamarthi et al., 2021) and (2) **DEEPAR** (Salinas et al., 2020) as state-of-the-art deep probabilistic forecasting models which do not exploit hierarchy relations. (3) **MINT** (Wickramasuriya et al., 2019) and (4) **ERM** (Ben Taieb & Koo, 2019) are methods that convert incoherent forecasts as post-processing step by framing it as an optimization problem. Since TSFNP provided better evaluation scores compared to DEEPAR, we performed ERM and MINT on Monte Carlo samples of TSFNP predictive distribution. (5) **HIERE2E** (Rangapuram et al., 2021) is a recent state-of-the-art deep-learning based approach that projects the raw predictions onto a space of coherent forecasts and trains the model in an end-to-end manner. (6) **SHARQ** (Han et al., 2021) is another state-of-the-art deep learning based approach that reconciles forecast distributions by using quantile regressions and making the quantile values coherent. (7) **PEMBU** (Taieb et al., 2017) is a post-processing method that refines raw forecasts to be distributionally coherent. We use the mean forecast from MINT and ERM as input forecasts for PEMBU. Note that we fine-tune the hyperparameters of PROFHIT and each baseline specific to each benchmark. More details on hyperparameters are in Appendix.

We also evaluate the efficacy and contribution of our various modeling choices by performing an ablation study using the following variants of PROFHIT: (7) **P-GLOBAL:** We study the effect of our multi-tasking hard-parameter sharing approach (Section 3.4) by training a variant where all the parameters are shared across all the nodes. (8) **P-FINETUNE:** We also look at the efficacy of

---

[2]Code and datasets: `https://anonymous.4open.science/r/PROFHiT-6F2F`

our soft regularization using both losses that adapts to optimize for both coherency and training accuracy by comparing it with a variant where the predictive distribution decoder parameters are further fine-tuned for individual nodes using only the likelihood loss. (9) **P-DEEPAR:** We evaluate our choice of using TSFNP, a previous state-of-the-art univariate forecasting model for accurate and calibrated forecasts with DeepAR, another popular probabilistic forecasting model that was used by HIERE2E. (10) **P-NOCOHERENT:** This variant is trained by completely removing the SDCR from the training. Note that unlike P-FINETUNE which was initially trained with SDCR before fine-tuning, P-NOCOHERENT never uses the SDCR at any point of training routine. Therefore P-NOCOHERENT measures the importance of explicitly regularizing over the information from the hierarchy.

**Datasets:** We evaluate on a diverse set of publicly available datasets (Table 2) from different domains with varied hierarchical relations and consistency. The bechmarking dataset and evaluation setup including forecast horizon is replicated from recent and past literature related to general HTSF

Table 2: Dataset Characteristics and Consistency

| Dataset | No. of Nodes | Levels of Hierarchy | $\tau$ | Obs. per node | Consistency (CE) |
|---|---|---|---|---|---|
| Tourism-L | 555 | 4,5 | 12 | 228 | Strong(0) |
| Labour | 57 | 4 | 8 | 514 | Strong(0) |
| Wiki | 207 | 5 | 1 | 366 | Strong(0) |
| Flu-Symptoms | 61 | 3 | 4 | 544 | Weak(3.37) |
| FB-Survey | 61 | 3 | 4 | 257 | Weak(2.44) |

as well as epidemic forecasting. (1) Labour dataset contains monthly employment data from Feb 1978 to Dec 2020 collected from Australian Bureau of Statistics. (2) Tourism-L (Wickramasuriya et al., 2019) contains tourism flows in different regions in Australia grouped via region and demographic. It has two sets of hierarchy (with four and five levels), one for the mode of travel and the other for geography with the top node being the only common node of both hierarchies. (3) Wiki dataset collects the number of daily views of 145000 Wikipedia articles aggregated into 150 groups (Taieb et al., 2017). These 150 groups are leaf nodes of a four-level hierarchy with groups of similar topics aggregated together. (4) Flu-Symptoms contains flu incidence values called *weighted influenza-like incidence* (wILI) values (Reich et al., 2019) at multiple spatial scales for USA for period of 2004-2020. The scales used are states, HHS and National level (US states are grouped into 10 HHS regions by CDC). (5) FB-Survey provides aggregated anonymized daily indicator for the prevalence of Covid-19 symptoms based on online surveys conducted on Facebook (Delphi Research Group, 2021) from Dec 2020 to Aug 2021 for each state and national level. We use the state-level values to find aggregates at HHS levels.

Tourism-L, Labour and Wiki are constructed by collecting values of leaf nodes and deriving the values of time-series of other nodes of the hierarchy. Hence, they are strongly consistent with zero CE (Definition 1). The values of each node of hierarchy in case of Flu-Symptoms and FB-Survey are directly collected or measured. For example, the values of Flu-Symptoms dataset are collected from public health agencies at the state, HHS and national levels and aggregated by CDC. Due to factors like reporting discrepancies and noise they contain values in time-series that may deviate from the given hierarchical relations (Chakraborty et al., 2018). Therefore, these datasets are weakly consistent with significant CE (Table 2).

**Evaluation metrics** For a ground truth $y^{(t)}$, let the predicted probability distribution be $\hat{p}_{y^{(t)}}$ with mean $\hat{y}^{(t)}$. Also let $\hat{F}_{y^{(t)}}$ be the CDF. We evaluate our model and baselines using carefully chosen metrics that are widely used in literature to measure accuracy and calibration. **1. Mean Absolute Percentage Error (MAPE)** is a commonly used score for point-predictions calculated as $MAPE = \frac{1}{N} \sum_{t=t_1}^{t_N} |\frac{y^{(t)} - \hat{y}^{(t)}}{y^{(t)}}|$ **3. Log Score (LS)** is a standard score used to measure accuracy of probabilistic forecasts in epidemiology (Reich et al., 2019). LS measures the negative log likelihood of a fixed size interval around the ground truth under the predictive distribution: $LS(\hat{p}_y, y) = -\int_{y-L}^{y+L} \log \hat{p}_y(\hat{y}) d\hat{y}$. Similar to (Reich et al., 2019), log likelihood of a forecast is capped at -10. **4. Calibration Score (CS):** To measure calibration of forecasts, we use the calibration score defined in Section 2. **2. Cumulative Ranked Probability Score (CRPS)** is a widely used standard metric for evaluation of probabilistic forecasts that measures *both accuracy and calibration*. Given ground truth $y$ and the predicted probability distribution $\hat{p}_y$, let $\hat{F}_y$ be the CDF. Then, CRPS is defined as: $CRPS(\hat{F}_y, y) = \int_{-\infty}^{\infty} (\hat{F}_y(\hat{y}) - \mathbf{1}\{\hat{y} > y\})^2 d\hat{y}$. We approximate $\hat{F}_y$ as a Gaussian distribution formed from samples of model to derive CRPS. **5. Distributional Coherency Error (DCE):** We calculate the Distributional Coherency Error (Equation 11) on output forecast distributions during inference to study how PROFHIT and baselines leverage SDCR to learn from hierarchical relations across datasets of varying consistency and trade-off coherency, calibration and accuracy, especially for weakly consistent data (Section 4.2 Q3).

## 4.2 RESULTS

We comprehensively evaluate PROFHIT through the following questions: **Q1:** Does PROFHIT predict accurate calibrated forecasts? **Q2:** Does PROFHIT provide consistently better performance across all levels of the hierarchy? **Q3:** Does SDCR help PROFHIT outperform baselines on both strongly and weakly consistent datasets? **Q4:** How does improved calibration affect robustness of PROFHIT's forecasts?

Table 3: Average scores (across 5 runs) across all levels of hierarchy for all baselines, PROFHIT and its variants. PROFHIT provides 54% better accuracy and 32% better calibration.

| Models/Data | Tourism-L | | | | | Labour | | | | | Wiki | | | | |
|---|---|---|---|---|---|---|---|---|---|---|---|---|---|---|---|
| | MAPE% | CRPS | LS | CS | DCE | MAPE% | CRPS | LS | CS | DCE | MAPE% | CRPS | LS | CS | DCE |
| DEEPAR | 3.12 | 0.17 | 0.61 | 0.19 | 0.32 | 18.27 | 0.045 | 0.75 | 0.25 | 0.34 | 16.52 | 0.232 | 0.83 | 0.27 | 0.26 |
| TSFNP | 2.28 | 0.21 | 1.19 | 0.14 | 0.39 | 14.52 | 0.071 | 1.41 | 0.21 | 0.22 | 15.63 | 0.287 | 0.86 | 0.21 | 0.39 |
| TSFNP-MinT | 1.17 | 0.5 | 0.58 | 0.15 | 0.24 | 16.46 | 0.045 | 4.12 | 0.26 | 0.12 | 13.79 | 0.243 | 0.78 | 0.18 | 0.18 |
| TSFNP-ERM | **1.42** | 0.56 | 0.53 | 0.11 | 0.18 | 13.57 | 0.045 | 3.63 | 0.23 | 0.19 | 17.74 | 0.221 | 0.74 | 0.19 | 0.21 |
| HIERE2E | 1.67 | 0.15 | 0.38 | 0.17 | 0.21 | **12.53** | 0.034 | 0.51 | 0.25 | 0.15 | 17.05 | 0.211 | 0.46 | 0.23 | 0.12 |
| SHARQ | 1.63 | 0.17 | 0.41 | 0.12 | 0.13 | 14.21 | 0.054 | 0.47 | 0.18 | 0.09 | 16.13 | 0.241 | 0.52 | 0.16 | 0.16 |
| PEMBU-MINT | 1.77 | 0.15 | 0.46 | 0.24 | 0.03 | 13.55 | 0.039 | 0.56 | 0.22 | 0.11 | 14.66 | 0.279 | 0.58 | 0.21 | 0.05 |
| PEMBU-ERM | 1.63 | 0.16 | 0.43 | 0.21 | **0.02** | 13.19 | 0.042 | 0.61 | 0.25 | **0.03** | 15.79 | 0.268 | 0.54 | 0.18 | **0.02** |
| PROFHIT | 1.47 | **0.12** | **0.33** | **0.09** | **0.02** | 12.79 | **0.026** | **0.21** | **0.14** | 0.05 | **12.47** | **0.184** | **0.35** | **0.13** | 0.04 |
| P-FINETUNE | 1.52 | 0.16 | 0.39 | 0.14 | 0.25 | 14.36 | 0.031 | 0.36 | 0.21 | 0.13 | 13.22 | 0.216 | 0.39 | 0.21 | 0.08 |
| P-GLOBAL | 1.47 | 0.13 | 0.42 | **0.06** | **0.01** | **12.17** | 0.027 | 0.31 | 0.16 | 0.04 | 12.37 | 0.185 | 0.34 | 0.16 | 0.04 |
| P-DEEPAR | **1.45** | 0.13 | 0.52 | 0.12 | 0.04 | 13.44 | 0.029 | 0.58 | 0.17 | 0.08 | 12.89 | 0.201 | 0.48 | 0.24 | 0.07 |
| P-NOCOHERENT | 1.82 | 0.18 | 0.37 | 0.21 | 0.35 | 16.97 | 0.043 | 0.45 | 0.26 | 0.17 | 17.44 | 0.227 | 0.47 | 0.35 | 0.14 |

| Models/Data | Flu-Symptoms | | | | | FB-Survey | | | | |
|---|---|---|---|---|---|---|---|---|---|---|
| | MAPE% | CRPS | LS | CS | DCE | MAPE% | CRPS | LS | CS | DCE |
| DEEPAR | 31.27 | 0.610 | 3.25 | 0.065 | 0.31 | 17.39 | 7.32 | 5.32 | 0.17 | 0.29 |
| TSFNP | 12.8 | 0.460 | 0.93 | 0.034 | 0.42 | 15.35 | 5.53 | 7.84 | 0.11 | 0.37 |
| TSFNP-MinT | 10.56 | 0.630 | 3.18 | 0.082 | 0.18 | 12.24 | 5.39 | 6.35 | 0.14 | 0.24 |
| TSFNP-ERM | 11.85 | 0.620 | 2.75 | 0.075 | 0.12 | 13.16 | 6.14 | 4.23 | 0.12 | 0.19 |
| HIERE2E | 15.67 | 0.420 | 0.81 | 0.12 | 0.32 | 12.63 | 4.12 | 1.13 | 0.19 | 0.26 |
| SHARQ | 18.34 | 0.470 | 1.42 | 0.071 | 0.21 | 12.82 | 3.12 | 0.81 | 0.15 | 0.19 |
| PEMBU-MinT | 15.44 | 0.621 | 2.55 | 0.18 | **0.05** | 13.75 | 5.78 | 4.22 | 0.22 | **0.07** |
| PEMBU-ERM | 17.57 | 0.688 | 2.74 | 0.15 | 0.07 | 12.99 | 6.31 | 5.18 | 0.18 | 0.1 |
| PROFHIT | 8.85 | **0.250** | **0.28** | **0.042** | 0.14 | 9.67 | **1.43** | **0.45** | **0.08** | 0.16 |
| P-FINETUNE | 10.44 | 0.240 | 0.3 | **0.039** | 0.17 | 9.83 | 1.18 | 0.72 | **0.07** | 0.19 |
| P-GLOBAL | 14.27 | 0.350 | 0.47 | 0.086 | 0.09 | 12.11 | 2.64 | 1.39 | 0.14 | 0.11 |
| P-DEEPAR | 17.43 | 0.361 | 0.54 | 0.083 | 0.15 | 11.89 | 2.13 | 0.75 | 0.18 | 0.15 |
| P-NOCOHERENT | 9.17 | 0.248 | 0.36 | 0.16 | 0.22 | 13.99 | 1.17 | 0.84 | 0.24 | 0.22 |

**Accuracy and calibration performance (Q1)**   We evaluate all baselines, PROFHIT and its variants for all the datasets over 5 independent runs. The average scores across all levels hierarchy are shown in Tables 3. PROFHIT significantly outperforms all baselines in MAPE score by 13% and LS by 14%-550%. In terms of calibration, we observe an average of 32% lower CS scores. Finally, PROFHIT shows 41-88% better CRPS scores. Thus, PROFHIT adapts well to varied kinds of datasets and outperforms all baselines in both accuracy and calibration. Performing t-test with significance $\alpha = 1\%$ we find that all the CRPS, LS and CS scores are statistically significant compared to baselines. On comparing the performance of PROFHIT with the variants, PROFHIT is comparable to or better than the best-performing variant in most benchmarks. This shows that all the important model design choices (multi-task parameter sharing, distributional coherency, and joint training on both losses) of PROFHIT are important for its consistently superior performance.

**Performance across the hierarchy (Q2)**   Next, we look at the performance of all models across each level of hierarchy. We compared the performance of PROFHIT with best performing baselines HIERE2E and SHARQ for all datasets. PROFHIT significantly outperforms the best baselines. At the leaf nodes, which contain most data, PROFHIT outperforms best baselines by 7% in `Wiki` to 100% in `FB-Survey`. For the top node of time-series the performance improvement is largest at 35% (`Wiki`) to 962% (`FB-Survey`). Similarly, for calibration score, we observe an average improvement of 12% for top nodes and 18% for bottom nodes. We show detailed results in Appendix. PROFHIT also performs better than the variants in most higher levels of hierarchy and its performance is comparable to the best variant (P-FINETUNE and P-GLOBAL) at leaf nodes as well. P-NOCOHERENT performs most poorly compared to all variant and PROFHIT, proving that SDCR is a very important contributor for consistent performance across the hierarchy in all datasets.

**SDCR leads to consistently better performance across varying data consistency (Q3)**   As discussed in Section 4.2, we evaluated on both strong and weakly consistent datasets. Since most previous state-of-the-art models assume datasets to be strongly consistent, deviations from this assumptions can cause under-performance when used with weakly consistent datasets. This is evidenced in Table 3 where most of the baselines explicitly optimize for hierarchical coherency

as a hard constraint on the forecasts. For example, PEMBU's forecasts have better distributional coherency error (DCE) for weakly consistent datasets. However, they perform much worse in both accuracy and calibration than even TSFNP, which does not even leverage hierarchical relations. Since we use SDCR as soft learning constraint, PROFHIT can learn to trade-off coherency for accuracy and calibration. Therefore, PROFHIT provides 93% better CRPS and 33% better calibration scores over best HTSF baselines. These improvements are more pronounced at non-leaf nodes of hierarchy where PROFHIT improves by 2.8 times for Flu-Symptoms and 9.2 times for FB-Survey. In case of strongly consistent datasets, PROFHIT provides 54% better CRPS and 23% better calibration scores while having comparable DCE to PEMBU. We observed that soft coherency regularization and parameter sharing across nodes are vital for PROFHIT's adaptability to varying levels of consistency. We provide detailed analysis of these observations in the Appendix.

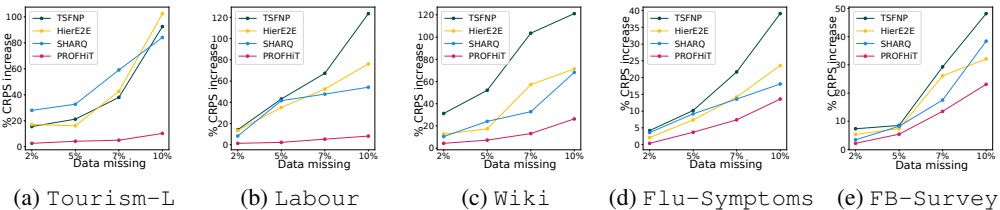

(a) Tourism-L  (b) Labour  (c) Wiki  (d) Flu-Symptoms  (e) FB-Survey

Figure 3: % increase in CRPS for all models with increase in proportion of missing data.

**Calibration enables robustness (Q4)** Accurate and well-calibrated models that can effectively leverage knowledge of the hierarchy can intuitively allow models to better adapt to noise/missing data. Hence we introduce the task of *Hierarchical Forecasting with Missing Values* to study the robustness of models when there are missing values in time-series. We model a situation that is encountered in many real-world applications such as Epidemic Forecasting where the past few values of time-series are missing due to various factors like data reporting delays (Chakraborty et al., 2018).

Formally, at time-period $t$, we are given full data up to time $t - \rho$. We set $\rho = 5$ since it is the average forecast horizon of all datasets. For sequence values in time period between $t - \rho$ and $t$, we randomly remove $k\%$ of these values across all time-series. The models are trained on complete time-series dataset till time $t' = t - \rho$. Models' predictions are then used to fill in missing values for time $t'$ to $t$. Finally, we input the filled time-series to generate forecasts for the future time-steps.

We measure relative decrease in performance with increase in percentage of missing data $k$ (Figures 3). We observe that PROFHIT's performance decrease with a larger fraction of missing values is much slower compared to other baselines. Even at $k = 10\%$, PROFHIT's performance decreases by 10.45-26.8% compared to other baselines that typically decrease by over 70%. Thus, PROFHIT effectively uses coherency to generate robust predictions on strong and weakly consistent datasets.

## 5 CONCLUSION AND DISCUSSION

We introduced PROFHIT, a probabilistic hierarchical forecasting model that produces accurate and well-calibrated forecasts using soft distributional coherency regularization (SDCR) which enables adaptablity to datasets with varying levels of hierarchical consistency. We evaluated PROFHIT against previous state-of-the-art hierarchical forecasting baselines over wide variety of datasets and observed 41-88% improvement average improvement in accuracy and 23-33% better calibration. PROFHIT provided best performance across the entire hierarchy as well as significantly outperformed other models in providing robust predictions when it encountered missing data where other baselines' performance degraded by over 70%.

Our work opens new possibilities like extending to various domains where time-series values across the hierarchy may not be continuous real numbers, can not be modelled as Gaussian distributions or may have different sampling rates. We can also explore modeling more complex structures between time-series with different aggregation relations. PROFHIT can also be used to study anomaly detection in time-series, especially in time-periods where there are deviations from assumed coherency relations. Similar to Kamarthi et al. (2022), we can extend our work to include multiple sources of features and modalities of data both specific to each time-series and global to the entire hierarchy.

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

# Appendix for PROFHIT: Probabilistic Robust Forecasting for Hierarchical Time-series

## A  ADDITIONAL RELATED WORK

**Probabilistic time-series forecasting** Classical probabilistic time-series forecasting methods include exponential smoothing and ARIMA (Hyndman & Athanasopoulos, 2018). They are simple but focus on univariate time-series and model each time-series sequence independently. Recently, deep learning based methods have been successfully applied in this area. DeepVAR (Salinas et al., 2020) trains an auto-regressive recurrent network model on a large number of related time series to directly output the mean and variance parameters of the forecast distribution. Other works are inspired from the space-state models and explicitly model the transition and emission components with deep learning modules such as deep Markov models (Krishnan et al., 2017) and deep state space models (Li et al., 2021; Rangapuram et al., 2018) Recently, EpiFNP (Kamarthi et al., 2021) has achieved state-of-art performance in epidemic forecasting. It learns the stochastic correlations between input data and datapoints to model a flexible non-parametric distribution for univariate sequences.

**Hierarchical time-series forecasting** Classical works on hierarchical time-series forecasting used a two-step approach (Hyndman et al., 2011; Hyndman & Athanasopoulos, 2018) and focus on point predictions. They first forecast for time-series only at a single level of the hierarchy and then derive the forecasts for other nodes using the hierarchical relations.

Recent methods like MINT and ERM are post-processing steps applied on the set of forecasts at all levels of hierarchy. MINT (Wickramasuriya et al., 2019; Wickramasuriya, 2021) assumes that the base level forecasts are uncorrelated and unbiased and solves an optimization problem to minimize the variance of forecast errors of past predictions. The unbiased assumption is relaxed in ERM (Ben Taieb & Koo, 2019). Corani et al. (2020) and Novak et al. (2017) uses a fully Bayesian bottom-up post-processing approach using raw forecasts from full hierarchy. Another line of works projects the raw forecasts of all time-series into a subspace of coherent forecasts. (Erven & Cugliari, 2015) use an iterative Game-theoretic approach of minimizing forecast error and projection error. Taieb et al. (2017) uses copula method to refine raw forecasts to be distributionally coherent as a post-processing step. Recent neural methods perform end-to-end learning that enables the model to leverage hierarchical relations while forecasting. Rangapuram et al. (2021) use a deep-learning based end-to-end approach to directly train on the projected forecasts. SHARQ (Han et al., 2021) is another recent probabilistic deep-learning based method that uses quantile regression and regularizes for coherency at different quantiles of forecast distribution. However, unlike our approach, these end-to-end methods do not regularize for coherency over the entire distribution (Distributional Coherency) but only over fixed quantiles. Most of these methods also are not designed for cases where the hierarchical constraints are not always consistently followed.

## B  CODE AND DATASET

We evaluated all models on a system with Intel 64 core Xeon Processor with 128 GB memory and Nvidia Tesla V100 GPU with 32 GB VRAM. We provide an anonymized repository of our implementation of PROFHIT along with the datasets used at `https://anonymous.4open.science/r/PROFHiT-6F2F`. We will release the code and data publicly after acceptance.

## C  HYPERPARAMETERS

### C.1  DATA PREPROCESSING

Most datasets used in our work assume the aggregation function to be simple summation (i.e, $\phi_{ij} = 1$ for all weights). We first normalize the values of leaf time-series training data to have 0 mean and variance of 1. Since the aggregation of values at higher levels of the hierarchy can lead to very large values in time-series, we instead divide each non-leaf time-series by the number of children. Then the weights of hierarchical relations become $\phi_{ij} = \frac{1}{|C_i|}$ where $C_i$ is the set of all children nodes of

time-series $i$. For the remaining datasets (Flu-Symptoms, FB-Symptoms) the time-series values are normalized by default and thus require no extra pre-processing.

## C.2 MODEL ARCHITECTURE

The architecture of TSFNP used in PROFHIT is similar to that used in the original implementation (Kamarthi et al., 2021). The GRU unit contains 60 hidden units and is bi-directional. Thus the local latent variable is also of dimension 60. $NN_1$ and $NN_2$ are both 2-layered neural networks with the first layer shared between both. Both layers have 60 hidden units. Finally, $NN_3$ is a three-layer neural network with the input layer having 180 units (for the concatenated input of three 60 dimensional vectors) and the last two layers having 60 hidden units. We found that the value of $c$ in Equation 9 is not very sensitive and usually set it to 5.

Note that we do not explicitly model covariance between every pair of time series (like MINT, ERM) and use a weighted combination of raw forecast parameters to derive refined forecasts. Therefore the refinement module complexity (Section 3.2) is $O(N^2)$ which is on par with previous methods like HIERE2E.

## C.3 TRAINING AND EVALUATION

Given the training dataset $\mathcal{D}_t$ we extract training dataset for each node as the set of prefix sequences $\{(\mathbf{y}_i^{(t1:t2)}, y_i^{(t2+1)}) : 1 \leq t1 \leq t2 < t - \tau\}$ and train the full model (TSFNP and refinement module). We tune the hyperparameter using backtesting by validating on window $t - \tau$ to $t$. Finally we train for entire training set with best hyperparameters.

For each benchmark, we used the validation set to mainly find the optimal batch size and learning rate. We searched over batch-size of $\{10, 50, 100, 200\}$ and the optimal learning rate was usually around 0.001. We also found the optimal $\lambda$ to be around 0.01 for strongly consistent datasets and 0.001 for weakly consistent datasets. We used early stopping with the patience of 150 epochs to prevent overfitting. For each independent run of a model, we initialized the random seeds from 0 to 5 for PyTorch and NumPy. We didn't observe large variations due to randomness for PROFHIT and all baselines.

During evaluation, we sampled 2000 Monte-Carlo samples of the forecast distribution and used it to estimate the mean for MAPE. We also used the samples mean and variance to evaluate LS and CS whereas used ensemble scoring to evaluate CRPS directly from the samples using `properscoring` package [3].

## D DERIVATION OF LIKELIHOOD ELBO LOSS

The full predictive distribution of PROFHIT from Equation 2 can be further expanded as:

$$P(\{y_i^{(t+\tau)}\}_{i=1}^N | \mathcal{D}_t) = \int \underbrace{\left( \prod_{i=1}^N P(\mathbf{u}_i | \mathbf{y}_i^{(1:t)}) \right)}_{\text{Probabilistic Encoder}} \underbrace{\left( \prod_{i=1}^N P(N_i | \{\mathbf{u}_i\}_{i=1}^N) P(\mathbf{z}_i | N_i) \right)}_{\text{SDCG}} \underbrace{P(\mathbf{z} | \{\mathbf{u}_i\}_{i=1}^N)}_{\text{Global Latent variable}}$$

$$\underbrace{\left( \prod_{i=1}^N P(\hat{\mu}_i, \hat{\sigma}_i | \mathbf{z}, \mathbf{z}_i, \mathbf{u}_i) \right)}_{\text{Raw forecasts}} \underbrace{\prod_{i=1}^N P(\mu_i, \sigma_i | \{\hat{\mu}_j, \hat{\sigma}_j\}_{j=1}^N) P(y_i^{(t+\tau)} | \mu_i, \sigma_i)}_{\text{Refinement Module}} d\{\mathbf{u}_i\}_{i=1}^N d\{\mathbf{z}_i\}_{i=1}^N.$$

$$(13)$$

To minimize the data likelihood $P(\{y_i^{(t+\tau)}\}_{i=1}^N | \mathcal{D}_t)$ requires intregration over latent variables $\{\mathbf{u}_i\}_{i=1}^N$ and $\mathbf{z}_i\}_{i=1}^N$. We instead perform amortized variational inference on the latent variables similar to VAE (Kingma & Welling, 2013).

---

[3] https://github.com/properscoring/properscoring

We approximate the posterior of latent variables $P(\{\mathbf{u}_i\}_{i=1}^N, \{\mathbf{z}_i\}_{i=1}^N, \{N_i\}_{i=1}^N, \mathbf{z}|\{y_i^{(t+\tau)}\}_{i=1}^N)$ with a variational distribution $Q(\{\mathbf{u}_i\}_{i=1}^N, \mathbf{z}_i\}_{i=1}^N, \{N_i\}_{i=1}^N, \mathbf{z}|\{y_i^{(t+\tau)}\}_{i=1}^N)$ expressed as:

$$Q(\{\mathbf{u}_i\}_{i=1}^N, \{\mathbf{z}_i\}_{i=1}^N, \{N_i\}_{i=1}^N, \mathbf{z}|\{y_i^{(t+\tau)}\}_{i=1}^N) = \left(\prod_{i=1}^N P(\mathbf{u}_i|\mathbf{y}_i^{(1:t)})\right)\left(\prod_{i=1}^N P(N_i|\{\mathbf{u}_i\}_{i=1}^N)P(\mathbf{z}_i|N_i)\right)$$
$$\left(\prod_{i=1}^N q_\phi(\mathbf{z}_i|\mathbf{y}_i^{(1:t)})\right) P(\mathbf{z}|\{\mathbf{u}_i\}_{i=1}^N)$$

$$(14)$$

where $q_\phi$ is a feed-forward network over GRU embeddings of Probabilistic Neural Encoder that parameterizes to a gaussain distribution of $\mathbf{z}_i$.

The ELBO loss

$$\mathbb{E}_{Q(\{\mathbf{u}_i, \mathbf{z}_i, N_i\}_{i=1}^N, \mathbf{z}|\{y_i^{(t+\tau)}\}_{i=1}^N)}[\log P(\{y_i^{(t+\tau)}\}_{i=1}^N|\{\mathbf{u}_i, \mathbf{z}_i, N_i\}_{i=1}^N, \mathbf{z})$$
$$+ \log P(\{\mathbf{u}_i\}_{i=1}^N, \{\mathbf{z}_i\}_{i=1}^N, \{N_i\}_{i=1}^N, \mathbf{z}|\{y_i^{(t+\tau)}\}_{i=1}^N) - \log Q(\{\mathbf{u}_i\}_{i=1}^N, \{\mathbf{z}_i\}_{i=1}^N, \{N_i\}_{i=1}^N, \mathbf{z}|\{y_i^{(t+\tau)}\}_{i=1}^N)]$$

$$(15)$$

get simplified to Equation 10 by cancelling of similar terms between variational and true distribution of latent variables.

## E   DERIVATION OF DISTRIBUTIONAL COHERENCY ERROR

*The Distributional Coherency Error (Equation 11) can be exactly expressed as:*

$$\mathcal{L}_2 = \sum_{i=1}^N \frac{\sigma_i^2 + \left(\mu_i - \sum_{j\in C_i}\phi_{ij}\mu_j\right)^2}{2\sum_{j\in C_i}\phi_{ij}^2\sigma_j^2} + \sum_{i=1}^N \frac{\sum_{j\in C_i}\phi_{ij}^2\sigma_j^2 + \left(\mu_i - \sum_{j\in C_i}\phi_{ij}\mu_j\right)^2}{2\sigma_i^2}. \quad (16)$$

To derive Equation 12, we use the following well-known result for JSD of two Gaussian Distributions (Nielsen, 2019):

**Lemma 1.** *Given two univariate Normal distributions $P_1 = \mathcal{N}_1(\mu_1, \sigma_1)$ and $P_2 = \mathcal{N}_2(\mu_2, \sigma_2)$, the JSD is*

$$JSD(P_1, P_2) = \frac{1}{2}\left[\frac{\sigma_1^2 + (\mu_1 - \mu_2)^2}{2\sigma_2^2} + \frac{\sigma_2^2 + (\mu_1 - \mu_2)^2}{2\sigma_1^2} - 1\right] \quad (17)$$

Consider each JSD term

$$\text{JSD}\left(P(y_i^{t+\tau}|\hat{\mu}_i, \hat{\sigma}_i), P\left(\sum_{j\in\mathbf{C}_i}\phi_{ij}y_j^{t+\tau}|\{|\hat{\mu}_j, \hat{\sigma}_j\}_{j\in\mathbf{C}_i}\right)\right) + 1$$

of the summation in Equation 11. Note that

$$P(y_i^{t+\tau}|\hat{\mu}_i, \hat{\sigma}_i) = \mathcal{N}(\mu_i, \sigma_i) \quad (18)$$

and $P(\sum_{j\in\mathbf{C}_i}\phi_{ij}y_j^{t+\tau}|\{|\hat{\mu}_j, \hat{\sigma}_j\}_{j\in\mathbf{C}_i}))$ is weighted sum of Gaussian variables $\{\mathcal{N}(\mu_j, \sigma_j)\}_{j\in C_i}$. Therefore,

$$P\left(\sum_{j\in\mathbf{C}_i}\phi_{ij}y_j^{t+\tau}|\{|\hat{\mu}_j, \hat{\sigma}_j\}_{j\in\mathbf{C}_i}\right) = \mathcal{N}\left(\sum_{j\in C_i}\phi_{ij}\mu_j, \sqrt{\sum_{j\in C_i}\phi_{ij}^2\sigma_j^2}\right). \quad (19)$$

Using Lemma 1 along with Equations 18,19 we get the desired result in Equation 16.

Table 4: *Average deviation of observed values in time-series from hierarchical relations.*

| Data | Flu | FB-Survey | Tourism-L | Labour | Wiki |
|---|---|---|---|---|---|
| Level 1 | 0.043 | 1.27 | 0 | 0 | 0 |
| Level 2 | 3.41 | 2.83 | 0 | 0 | 0 |
| Average across hierarchy | 3.37 | 2.44 | 0 | 0 | 0 |

## F  CONSISTENCY OF DATASETS

We noted in Section 4.2 Q4 that `Flu-Symptoms` and `FB-Survey` are weakly consistent datasets since they do not strictly follow the aggregation relations $H_{\mathcal{T}}$ unlike strongly consistent datasets `Tourism-L, Labour, Wiki`.

We empirically observe this by measuring Consistency errors of all datasets (Definition 1) for entire hierarchy and at each level of the hierarchy. The results are in Table 4. As expected there is no deviations for strongly consistent datasets where as there is significant deviation in weakly consistent data.

Table 5: *Average CRPS scores at each level of hierarchy.* PROFHIT *significantly outperforms best baselines across all benchmarks. Note that P-Finetune's performance decreases at higher levels of hierarchy compared to other variants whereas P-Global's performance is worse at lower levels.*

| Models/Data | Tourism-L | | | | | | | | Labour | | | |
|---|---|---|---|---|---|---|---|---|---|---|---|---|
| Hierarchy Levels | 1 | 2(Travel) | 3(Travel) | 4(Travel) | 5(Travel) | 2(Geo) | 3(Geo) | 4(Geo) | 1 | 2 | 3 | 4 |
| HIERE2E | 0.081 | 0.103 | 0.141 | 0.205 | 0.272 | 0.103 | 0.136 | 0.175 | 0.031 | 0.034 | 0.034 | 0.038 |
| SHARQ | 0.093 | 0.131 | 0.163 | 0.218 | 0.295 | 0.131 | 0.138 | 0.152 | 0.097 | 0.124 | 0.133 | 0.149 |
| PEMBU-MINT | 0.112 | 0.121 | 0.139 | 0.203 | 0.185 | 0.116 | 0.128 | 0.167 | 0.063 | 0.033 | 0.042 | 0.085 |
| PROFHIT (Ours) | **0.051** | **0.095** | **0.12** | **0.17** | **0.264** | **0.083** | **0.106** | **0.142** | **0.023** | **0.019** | **0.023** | **0.029** |
| P-FINETUNE | 0.072 | 0.136 | 0.083 | 0.16 | 0.278 | 0.124 | 0.124 | 0.158 | 0.024 | 0.022 | 0.026 | 0.035 |
| P-GLOBAL | 0.093 | 0.113 | 0.122 | 0.13 | **0.261** | 0.093 | 0.113 | 0.147 | **0.021** | 0.027 | 0.028 | **0.027** |
| P-DEEPAR | 0.075 | 0.097 | 0.136 | 0.183 | 0.281 | 0.095 | 0.122 | 0.159 | 0.025 | 0.027 | 0.031 | 0.033 |
| P-NOCOHERENT | 0.086 | 0.142 | 0.107 | 0.18 | 0.265 | 0.132 | 0.138 | 0.147 | 0.027 | 0.031 | 0.029 | 0.026 |
| Models/Data | Wiki | | | | | Flu-Symptoms | | | FB-Survey | | | |
| Hierarchy Levels | 1 | 2 | 3 | 4 | 5 | 1 | 2 | 3 | 1 | 2 | 3 | |
| HIERE2E | 0.042 | 0.105 | 0.229 | 0.272 | 0.372 | 0.272 | 0.421 | 0.458 | 4.14 | 4.04 | 4.13 | |
| SHARQ | 0.039 | 0.136 | 0.235 | 0.291 | 0.378 | 0.258 | 0.376 | 0.381 | 3.08 | 3.21 | 3.13 | |
| PEMBU-MINT | 0.031 | 0.171 | 0.241 | 0.385 | 0.433 | 0.337 | 0.567 | 0.773 | 4.82 | 5.53 | 6.15 | |
| PROFHIT (Ours) | **0.031** | **0.074** | **0.133** | **0.216** | **0.252** | **0.216** | **0.133** | **0.338** | **0.32** | **0.43** | **1.89** | |
| P-FINETUNE | 0.034 | 0.086 | 0.153 | 0.232 | 0.275 | 0.222 | 0.175 | **0.293** | 0.43 | 0.65 | **1.83** | |
| P-GLOBAL | 0.048 | 0.103 | 0.187 | 0.265 | **0.186** | 0.269 | 0.213 | 0.376 | 0.37 | **0.37** | 2.11 | |
| P-DEEPAR | 0.035 | 0.094 | 0.193 | 0.251 | 0.285 | 0.242 | 0.217 | 0.328 | 0.44 | 0.61 | 2.01 | |
| P-NOCOHERENT | 0.49 | 0.117 | 0.93 | 0.258 | 0.167 | 0.227 | 0.193 | 0.381 | 0.42 | **0.36** | 2.18 | |

Table 6: *Average CS scores at each level of hierarchy.* PROFHIT *significantly outperforms best baselines across all benchmarks.*

| Models/Data | Tourism-L | | | | | | | | Labour | | | |
|---|---|---|---|---|---|---|---|---|---|---|---|---|
| Hierarchy Levels | 1 | 2 | 3 | 4 | 5 | 2(Geo) | 3(Geo) | 4(Geo) | 1 | 2 | 3 | 4 |
| HIERE2E | 0.15 | 0.18 | 0.17 | 0.21 | 0.24 | 0.19 | 0.18 | 0.22 | 0.21 | 0.23 | 0.22 | 0.27 |
| SHARQ | 0.09 | 0.08 | 0.12 | 0.11 | 0.14 | 0.11 | 0.12 | 0.16 | 0.16 | 0.16 | 0.15 | 0.21 |
| PEMBU-MINT | 0.14 | 0.21 | 0.22 | 0.21 | 0.26 | 0.18 | 0.23 | 0.25 | 0.21 | 0.22 | 0.24 | 0.21 |
| PROFHIT | **0.05** | **0.06** | **0.04** | **0.06** | **0.11** | **0.06** | **0.06** | **0.1** | **0.17** | **0.11** | **0.15** | **0.16** |
| P-FINETUNE | 0.09 | 0.12 | 0.13 | 0.17 | 0.13 | 0.11 | 0.13 | 0.15 | 0.24 | 0.21 | 0.24 | 0.22 |
| P-GLOBAL | 0.06 | **0.04** | **0.03** | 0.08 | **0.05** | **0.05** | **0.03** | **0.04** | **0.14** | 0.18 | 0.19 | **0.15** |
| P-DEEPAR | 0.11 | 0.09 | 0.09 | 0.14 | 0.13 | 0.15 | 0.14 | 0.13 | 0.14 | 0.19 | 0.17 | 0.14 |
| P-NOCOHERENT | 0.18 | 0.19 | 0.17 | 0.19 | 0.22 | 0.18 | 0.19 | 0.24 | 0.24 | 0.22 | 0.25 | 0.31 |
| Models/Data | Wiki | | | | | Flu-Symptoms | | | FB-Survey | | | |
| Hierarchy Levels | 1 | 2 | 3 | 4 | 5 | 1 | 2 | 3 | 1 | 2 | 3 | |
| HIERE2E | 0.15 | 0.21 | 0.26 | 0.22 | 0.24 | 0.11 | 0.13 | 0.11 | 0.21 | 0.19 | 0.18 | |
| SHARQ | 0.13 | 0.14 | 0.14 | 0.17 | 0.15 | 0.58 | 0.052 | 0.085 | 0.16 | 0.14 | 0.15 | |
| PEMBU-MINT | 0.12 | 0.11 | 0.12 | 0.13 | 0.14 | 0.17 | 0.22 | 0.17 | 0.2 | 0.19 | 0.16 | |
| PROFHIT | **0.11** | **0.15** | **0.12** | **0.14** | **0.11** | **0.031** | **0.044** | **0.052** | **0.09** | **0.07** | **0.06** | |
| P-FINETUNE | 0.19 | 0.18 | 0.23 | 0.22 | 0.24 | 0.033 | **0.031** | 0.042 | **0.05** | **0.06** | 0.09 | |
| P-GLOBAL | 0.16 | **0.15** | 0.16 | 0.17 | 0.15 | 0.065 | 0.072 | 0.096 | 0.11 | 0.13 | 0.17 | |
| P-DEEPAR | 0.21 | 0.24 | 0.26 | 0.22 | 0.23 | 0.064 | 0.077 | 0.083 | 0.15 | 0.19 | 0.17 | |
| P-NOCOHERENT | 0.29 | 0.28 | 0.35 | 0.33 | 0.37 | 0.22 | 0.18 | 0.14 | 0.22 | 0.25 | 0.21 | |

# H  Details on Data Imputation Experiment

**Motivation:**  During real-time forecasting in real-world applications such as Epidemic or Sales forecasting, we encounter situations where the past few values of time-series are missing or unreliable for some of the nodes. This is observed specifically at lower levels, due to discrepancies or delays during reporting and other factors (Chakraborty et al., 2018). Therefore, one approach to performing forecasting in such a situation is first by imputation of missing values based on past data and then using the predicted missing values as part of the input for forecasting.

**Task:**  To simulate such scenarios of missing data and evaluate the robustness of PROFHIT and all baselines, we design a task called *Hierarchical Forecasting with Missing Values* (HFMV). Formally, at time-period $t$, we are given full data for up to time $t - \rho$. We show results here for $\rho = 5$ which is the average forecast horizon of all tasks. For sequence values in time period between $t - \rho$ and $t$, we randomly remove $k\%$ of these values across all time-series. The goal of HFMV task is to use the given partial dataset from $t - \rho$ to $t$ as input along with complete dataset for time-period before $t - \rho$ to predict future values at $t + \tau$. Therefore, success in HFMV implies that models are robust to missing data from recent past by effectively leveraging hierarchical relations.

**Setup:**  We first train PROFHIT and baselines on complete dataset till time $t'$ and then fill in the missing values of input sequence using the trained model. Using the predicted missing values, we again forecast the output distribution. For each baseline and PROFHIT, we perform multiple iterations of Monte-Carlo sampling for missing values followed by forecasting future values to generate the forecast distribution. We estimate the evaluation scores using sample forecasts from all sampling iterations.

We compared the performance of PROFHIT with best performing baselines HIERE2E and SHARQ for each level of hierarchy of all datasets. PROFHIT significantly outperforms the best baselines as well as the variants. At the leaf nodes, which contains most data, PROFHIT outperforms best baselines by 7% in `Wiki` to 100% in `FB-Survey`. For the top node of time-series the performance improvement is largest at 35% (`Wiki`) to 962% (`FB-Survey`). We show detailed results in Table 5

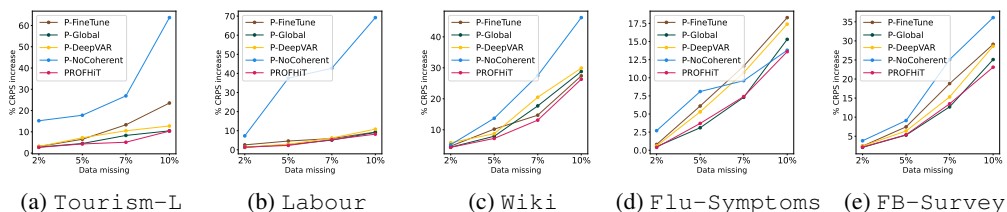

| (a) `Tourism-L` | (b) `Labour` | (c) `Wiki` | (d) `Flu-Symptoms` | (e) `FB-Survey` |

Figure 4: *% increase in CRPS for* PROFHIT *and variants with increase in proportion of missing data.*

**Robustness of PROFHIT variants:**  We compare relative performance decrease with increase in percentage of missing data for PROFHIT and its variants in Figure 4. We observe that P-NOCOHERENT's performance deteriorates very rapidly in most benchmarks, showing the importance of SDCR for learning provides robust calibrated coherent forecasts. The second worse-performing variant across all datasets is P-FINETUNE which also relies less on the hierarchical relations due to fine-tuning of parameters for specific time-series. Finally, we observe that PROFHIT and P-GLOBAL suffer the least degradation in performance since both these models prioritize integrating hierarchical coherency information which enables them to provide better estimates for imputed data for missing input and use them to generate more accurate and calibrated forecasts.

# I  Adapting to varying dataset consistency

**Observation 1.**  *The average improvement in performance of* PROFHIT *over best* `HTSF` *baselines is 72% higher for weakly consistent datasets over its improvement for strongly consistent datasets.*

Since most previous state-of-art models assume datasets to be strongly consistent, deviations to this assumptions can cause under-performance when used with weakly consistent datasets. This is evidenced in Table 3 where some of the baselines like MɪNT and ERM that explicitly optimize for hierarchical coherency perform worse than even TSFNP, which does not leverage hierarchical relations, in `Flu-Symptoms` and `FB-Survey`. Overall, we found that for weakly consistent datasets, PROFHɪT provides a much larger 93% average improvement in CRPS scores over best `HTSF` baselines compared to 54% average improvement for strongly consistent datasets. These improvements are more pronounced at non-leaf nodes of hierarchy where PROFHɪT improves by 2.8 times for `Flu-Symptoms` and 9.2 times for `FB-Survey`. This is because `HTSF` baselines which assume strong consistency do not adapt to noise at leaf nodes that compound to errors at higher levels of hierarchy.

**Observation 2.** PROFHɪT*'s approach to parameter sharing and soft coherency regularization helps adapt to varying hierarchical consistency.*

We observe that that best performing variant for strongly consistent datasets in P-GLOBAL which is trained with both likelihood loss and SDCR (Table 3). But its performance severely degrades for weakly consistent datasets since sharing all model parameters across all time-series makes it inflexible to model patterns and deviations specific to individual nodes. In contrast, P-FINETUNE and P-NOCOHERENT performs the best among variants for weakly consistent datasets since they train separate sets of decoder parameters for each node. But they perform poorly for strongly consistent datasets since they don't leverage Distributional Coherency effectively. PROFHɪT combines the flexible parameter learning of P-FINETUNE and leverage Distributional Coherency to jointly optimize the parameters like P-GLOBAL providing comparable performance to best variants over all datasets.

**Observation 3.** PROFHɪT*'s Refinement module automatically learns to adapt to varying hierarchical consistency.*

The design choices of the refinement module help PROFHɪT to adapt to datasets of different levels of hierarchical consistency. Specifically, by optimizing for values of $\{\gamma_i\}_{i=1}^N$ of Equation 8, PROFHɪT aims to learn a good trade-off between leveraging prior forecasts for a time-series and hierarchical relations of forecasts from the entire hierarchy. We study the learned values of $\{\gamma_i\}_{i=1}^N$ of Equation 8 used to derive refined mean. Note that higher values of $\gamma_i$ indicate larger dependence on raw forecasts of node and smaller dependence of forecasts of the entire

Table 7: Average value of $\gamma_i$ for all datasets. Note that weakly coherent datasets have higher $\gamma_i$ (depends mode on past data of same time-series) where as strongly-coherent data have lower $\gamma_i$ (leverages the hierarchical relations).

| Consistency | Dataset | Average value of $\gamma_i$ |
|---|---|---|
| Strong | Tourism-L | $0.420 \pm 0.096$ |
| | Labour | $0.348 \pm 0.091$ |
| | Wiki | $0.313 \pm 0.057$ |
| Weak | Symp | $0.759 \pm 0.152$ |
| | Fbsymp | $0.789 \pm 0.180$ |

hierarchy. We plot the average values of $\gamma_i$ for each of the datasets in Table 7. We observe that strongly consistent datasets have lower values of $\gamma_i$ indicating that PROFHɪT's refinement module automatically learns to strongly leverage the hierarchy for these datasets compared to weakly coherent datasets.

Table 8: **Std. dev** *of CRPS and LS (accros 5 runs) across all levels for all baselines,* PROFHɪT *and its variants.* PROFHɪT *performs significantly better than all baselines as noted using t-test with* $\alpha = 1\%$.

| | Models/Data | Tourism-L | | Labour | | Wiki | | Flu-Symptoms | | FB-Survey | |
|---|---|---|---|---|---|---|---|---|---|---|---|
| | | CRPS | LS | CRPS | LS | CRPS | LS | CRPS | LS | CRPS | LS |
| **Baselines** | DEEPAR | 0.011 | 0.040 | 0.004 | 0.038 | 0.002 | 0.044 | 0.018 | 0.098 | 0.482 | 0.434 |
| | TSFNP | 0.006 | 0.021 | 0.003 | 0.018 | 0.015 | 0.069 | 0.019 | 0.004 | 0.251 | 0.217 |
| | MɪNT | 0.005 | 0.019 | 0.002 | 0.121 | 0.018 | 0.006 | 0.014 | 0.111 | 0.468 | 0.213 |
| | ERM | 0.044 | 0.005 | 0.002 | 0.110 | 0.016 | 0.069 | 0.018 | 0.133 | 0.148 | 0.209 |
| | HɪERE2E | 0.001 | 0.038 | 0.003 | 0.049 | 0.019 | 0.018 | 0.005 | 0.051 | 0.325 | 0.109 |
| | SHARQ | 0.000 | 0.011 | 0.001 | 0.046 | 0.017 | 0.007 | 0.002 | 0.116 | 0.133 | 0.048 |
| | PROFHɪT (Ours) | 0.001 | 0.017 | 0.001 | 0.003 | 0.001 | 0.030 | 0.005 | 0.009 | 0.040 | 0.008 |
| **Ablation** | P-FINETUNE | 0.016 | 0.031 | 0.003 | 0.003 | 0.016 | 0.014 | 0.001 | 0.006 | 0.090 | 0.004 |
| | P-GLOBAL | 0.012 | 0.033 | 0.000 | 0.013 | 0.002 | 0.001 | 0.033 | 0.024 | 0.248 | 0.119 |
| | P-DEEPAR | 0.006 | 0.026 | 0.001 | 0.028 | 0.005 | 0.043 | 0.035 | 0.030 | 0.103 | 0.065 |
| | P-NOCOHERENT | 0.005 | 0.012 | 0.001 | 0.009 | 0.015 | 0.043 | 0.012 | 0.025 | 0.110 | 0.053 |

