# OpenReview forum: "When Rigid Coherency Hurts: Distributional Coherency Regularization for Probabilistic Hierarchical Time Series Forecasting"
_ICLR.cc/2023/Conference — Submitted to ICLR 2023_

### Official Review · Reviewer_UgsV · 2022-10-21

**Confidence:** 3
**Correctness:** 2
**Technical Novelty And Significance:** 3
**Empirical Novelty And Significance:** 3
**Recommendation:** 5

**Clarity, Quality, Novelty And Reproducibility:**

The paper is mostly clear written, I only have some minor comments:
* The author mentioned TSFNP is modified fromKamarthi et al., 2021, what is the difference and why? It’s provided in the appendix but it will help to also summarize it briefly in the main text.
* In eq(11), the loss is computed on the refined $\mu, \sigma$ or the raw $\hat{\mu}, \hat{\sigma}$?
* Typo in the “IS measures the negative log likelihood…” IS->LS.
* Why not make the hyperparameter $c$ in eq(9) learnable?
* What are the hyperparameters and training details for the baselines?

The overall writing and technical qualities are good. To the best of my knowledge, enforcing hierarchical constraints in the probabilistic space for forecasting is novel. Good reproducibility is backed up with implementation detail and code.


**Strength And Weaknesses:**

Strength:
* The paper is well motivated. It’s an Interesting setting with weakly consistent datasets, where the data generation process may follow a hierarchical set of constraints but may contain some deviations. HIERE2E (Rangapuram et al., 2021) and SHARQ (Han et al., 2021) impose hierarchical constraints on the mean or fixed quantiles of the forecast distributions, but the distribution may not be well calibrated.

* Convincing experiments. The work considered many relevant baselines and reported various metrics, characterizing different aspects. The 4 model variants provide data for the impact of the model choices. Replacing TSFNP with DeepAR and comparing to baselines with DeepAR separates out the effect of the base forecasting model.

On the weakness, the proposed method assumes Gaussian for the prediction, which is fair start but it limits the choices. Extending the framework to other distributions can be beneficial.

=== After discussion with AC and other reviewers ===

I think the idea is interesting and the empirical evaluation is quite comprehensive. But at the same time, the math behind the model especially on the ELBO, as pointed out by the AC, is alarming to me. There are steps missing and it is hard to verify the correctness of the derivations. Please see more detail in the above for this point. This negatively impacts the soundness of the work. Thus, I will lower my score accordingly.

**Summary Of The Paper:**

This work proposed a deep probabilistic framework for modeling the distributions of each time-series together using a soft distributional coherency regularization (SDCR) over forecast distributions. It has the advantages of 1) adapting to both strong and weak hierarchical consistent datasets and 2) producing well calibrated forecasting at all levels. The empirical results demonstrated that it improves performance on forecasting accuracy and calibration on all levels and provides robust predictions with missing data.


**Summary Of The Review:**

The paper is well motivated. It fills the gap of enforcing hierarchical constraints in the probabilistic space, allowing for soft matching and improved calibration. The chosen methodology makes sense to me and the empirical work is relatively strong and convincing. It has limitations in the Gaussian output, but as demonstrated in the results, it already performs strongly compared to the baselines. The paper is mostly clearly written and the code is also provided for reproducibility.

---

### Official Review · Reviewer_eMuv · 2022-10-24

**Confidence:** 4
**Correctness:** 2
**Technical Novelty And Significance:** 3
**Empirical Novelty And Significance:** 3
**Recommendation:** 5

**Clarity, Quality, Novelty And Reproducibility:**

This paper is technically sound and well-written. The authors have clearly distinguished the novelty over prior works and compared it with the baselines. They have also provided the code to reproduce the experiments.

**Strength And Weaknesses:**

Strength:
1. This paper proposed a fully parameterized and probabilistic framework for HTS forecasting, which flexibly incorporates hierarchical information into both point and distributional forecasts by enabling parameter sharing.
2. The proposed method obtains better performance on both point and probabilistic forecasts for various datasets.
3. The authors have distinguished strong and weak consistency, i.e., whether time series at a lower level should strictly add up to obtain time series at the upper level, which is a useful step for many real-world applications.

Weakness:
1. Since the authors have distinguished strong and weak consistency in the main paper, it is also better to separate (or at least highlight) them in the experiment since some methods are designed for strong consistency (e.g., MinT, ERM, and HIERE2E) and some method is better for weak consistency (e.g., SHARQ). It is not fair to compare these methods from different categories.
2. It seems that there is no formal guarantee of the robustness of PROFHIT over other baselines. The authors are using missing value imputation whereas other methods can also do. This experiment seems heuristic.
3. Normally in many applications, time series at different aggregation levels have distinct properties over sparsity, noise distribution or sampling rate etc., which will affect the distributional assumption at each level. Since the proposed method uses a general Gaussian distribution for each level, the authors could discuss how this general assumption can be adapted to datasets with different properties.

**Summary Of The Paper:**

This paper proposes a probabilistic hierarchical time series forecasting model that provides a coherent distributional forecast that achieves better performance in forecasting accuracy. The core idea is to use a deep probabilistic forecasting framework, i.e., the neural Gaussian process, to produce a probabilistic base forecast and then refine the base forecast using a separate module. This refinement procedure is based on a variational inference framework coupled with a distributional coherency loss. The proposed method has achieved improved performance over various baselines, particularly in datasets with missing values.

**Summary Of The Review:**

This paper proposed a novel method to address an important problem. I would recommend accepting this paper upon my concerns are addressed.

---

### Official Review · Reviewer_B3vd · 2022-10-25

**Confidence:** 5
**Correctness:** 2
**Technical Novelty And Significance:** 3
**Empirical Novelty And Significance:** 2
**Recommendation:** 1

**Clarity, Quality, Novelty And Reproducibility:**

Unfortunately, the paper lacks rigor and clarity.

The proposed method is novel, but it is not clear how it relates to existing methods, such as MINT, which makes similar assumptions.

Regarding reproducibility, the authors provide code to reproduce their experiments.

**Strength And Weaknesses:**

Strength
- More flexibility and robustness in probabilistic forecasting methods are important for many applications.
- The method can deal with inconsistent hierarchies.
- The proposed method provides good accuracy for various datasets.

Weaknesses

- The paper lacks rigor and clarity.

- Important related work is missing:
	- https://www.sciencedirect.com/science/article/abs/pii/S0377221722006087
	- https://arxiv.org/pdf/2103.11128.pdf
	- Bayesian methods
		- https://link.springer.com/chapter/10.1007/978-3-030-67664-3_13
		- https://arxiv.org/abs/1711.04738

- It is not clear how the proposed method relates to existing work, e.g. MINT. Given that the authors make a gaussianity assumption, it is not clear how their method compares with MINT, for which we have closed-form expression for the entire hierarchy (for means and covariance matrix).


- The logscore is improper with respect to incoherent predictive distributions. As a result, you cannot compare coherent with incoherent distributions with LS. See https://www.sciencedirect.com/science/article/abs/pii/S0377221722006087 for more details.

- The authors claim that their method provides well-calibrated forecasts. However, this concept it not clearly defined (See Definition 3). Please use your predictive model p_M in the definition.

- The authors claim that they model the joint distribution of all series in the hierarchy. However, in the sentence before Section 3, the authors refer to the marginals. This is not clear. Expression (2) gives the joint distribution of y_i for i=1, .., N, but the first sentence in "overview" talks about the marginal distributions.

- The authors should clearly explain the factorization in (1).

Comments:
- "We performed ERM and MINT on Monte Carlo samples of TSFNP predictive distribution": Please be more precise.

- The authors use both consistency and coherency. I would suggest sticking to coherency.

- CRPS does not measure accuracy and calibration, but sharpness and calibration.

- Various notions of "distributional coherency" have been proposed in the literature. A precise definition is needed here. I think you are using sample coherency

- "Samples of distribution": do you mean samples from the predictive distribution?

- In latex, please use \citep{} instead of \cite{}.

- What do you mean by "time-series of employment"?

- "both so-called strong and weakly consistent datasets": I am not sure "so-called" is the right term to use here.

- "They produce unreliable confidence intervals": I think you meant "prediction intervals".

- In section 2, please specify the range of phi_{i,j}. Does it belong to {0, 1}?

- Notations: \hat{p}(Y_t), \hat{y}_t, etc.

- Note that the CRPS has a closed-form expression for a Gaussian distribution.

- The authors should discuss the limitations of their method (computational complexity, statistical assumptions, etc)


**Summary Of The Paper:**

The paper proposes a new probabilistic forecasting method for hierarchical time series. Compared to existing methods, the proposed method is more robust and can deal with inconsistent hierarchies. Specifically, the method is based on a flexible Bayesian approach which includes distributional coherency regularization. Experiments on multiple datasets show that the proposed method improves accuracy compared to baselines and provides reliable forecasts even if up to 10% of input time-series data is missing.


**Summary Of The Review:**

I have reviewed this paper multiple times (for other conferences), and have given many suggestions for improvement. Unfortunately, the authors have ignored important past comments.

The biggest weakness of the paper is the lack of rigor and the quality of writing (notations, definitions, etc). Understanding differences and similarities with existing methods is also very important. Calibration is also not well defined.

Unless the writing is significantly improved, I do not see this paper being accepted at any top ML conference.

---

> ### Author Response · Authors · 2022-11-29
> **Summary of changes based on previous and current reviews**
>
> Dear reviewer B3vd,
>
> We wanted to summarize and highlight the changes we have made in the paper based on your suggestions in your current as well as previous reviews.
>
> ## Changes based on the reviewer’s earlier reviews
>
> 1. Added formal definitions for data consistency in Section 2 (see Definitions 1 and 2).
> 2. Added a formal measure of calibration via Calibration Score (CS) in Definition 3 which is based on past works [1,2].
> 3. Further to address the request for a more direct measure of calibration, we also added CS as an evaluation metric to our experiments and show that PROFHIT provided better-calibrated forecasts compared to other methods (Table 3).
> 4. Clarified the meaning of Distributional Coherency by providing a formal definition (Def. 4)
> 5. Added Distributional Coherency Error (DCE) as another evaluation metric to explicitly measure distributional coherency of forecasts from our model and baselines on various benchmarks (Table 3)
> 6. Clarified applying post-processing methods like ERM and MINT on TSFNP by reconciliation of Monte Carlo samples.
> 7. Provided derivation of Log-likelihood ELBO loss and Distributional coherency Loss in Appendix Sections D and E.
> 8. Added and contextualized PEMBU, a recent state-of-art two-step post-processing method for probabilistic hierarchical forecasting in the introduction and additional related work. PEMBU was also used as an additional baseline in our experiments.
>
> ## Changes based on the reviewer’s current review
>
> 1. Elaborated on the latent variables in Eq. 2 by adding :
> > where $\mathbf{z}_i, \mathbf{u}_i, \mathbf{z}$ are intermediate latent variables of our probabilistic raw forecasting model TSFNP (Section 3.1).
> 2. Replaced \cite{} with \citep{}
> 3. Added the possible values of $\phi_{i,j}$ in Section 2 (first paragraph) with phrase
> > $\phi_{i,j}$ are known and time-independent real-valued constants.
> 4. Added a note on the computational complexity of the refinement module in Appendix Section C.2 as described in review response
> 5. Added [3,4] to related works in Appendix Section A
>
> Thank you,
>
> Authors
>
> ## References
>
> [1] Kuleshkov et al. Accurate uncertainties for deep learning using calibrated regression ICML 2018
>
> [2] Kamarthi et al. Neural non-parametric uncertainty quantification for epidemic forecasting. NeurIPS 2021
>
> [3] Corani, Giorgio, et al. "Probabilistic reconciliation of hierarchical forecast via Bayes’ rule." Joint European Conference on Machine Learning and Knowledge Discovery in Databases. Springer, Cham, 2020.
>
> [4] Wickramasuriya, Shanika L. "Probabilistic forecast reconciliation under the Gaussian framework." arXiv preprint arXiv:2103.11128 (2021)

---

> ### Comment · Reviewer_cEHM · 2022-12-05
> **Minor comment about literature review**
>
> >  Some of the papers the reviewer pointed out ... hardly cited [13] (2 citations in 2 years according to Google Scholar).
>
> Not the original reviewer that suggested the reference, but I don't think a paper's lack of enough citations is a valid reason to further omit citing it. What gets into the self-sustaining citation cycle is dictated by an imperfect random process affected by social considerations, not some principled measure of scientific contribution. And I find it alarming that the authors find this to be justifiable grounds to omit a reference in a rebuttal.

---

### Official Review · Reviewer_cEHM · 2022-10-26

**Confidence:** 3
**Correctness:** 3
**Technical Novelty And Significance:** 3
**Empirical Novelty And Significance:** 4
**Recommendation:** 5

**Clarity, Quality, Novelty And Reproducibility:**

### Clarity

The paper is hard to read. The paper frequently introduces terms and concepts it only later defines. I understand there are several concepts involved and this might sometimes be hard. At the very least abbreviations should be expanded when stated first (e.g. TSFNP). Nevertheless, combined with the widespread vspace abuse and several notational issues, this makes reading the paper somewhat challenging overall.

#### Notational issues:
- Definition of H_T has "for every i in 1...N". However, the definition only applies to non-leaf nodes and therefore cannot apply to all N time series in the dataset.
- If the time variable is in the superscript, I suggest being consistent and writing D_t as D^t.
- What is u?
- In Eq. 4, if u is a vector, what is mu(u)? Seems like here u oughht to be a random variable not a vector.
- In Eq. 4, why is the subscript i italics in some places and roman in others?
- When N_i is a term explicitly used, please do not denote networks as NN_1, which simply adds to the confusion.
- Is it LS or IS? When defining the metric the paper says both LS and IS. For Q1 results, the paper says that the proposed method improves IS by up to 550% which seems like a typo.

#### Other writing improvement issues/suggestions:
- "rigid coherency on generated forecasts generate" -> remove last 'generate'
- Fix citation style, citations should be in parenthesis unless using \citet
- "Most state-of-art HTSF methods" -> "state-of-the-art"
- vspace abuse around the tables and figures.
- vspace abuse around equations (see top of Eq. 2)

### Quality

It is hard to evaluate the work rigorously given the writing and notational issues, and the lack of code and data in the supplementary material (the link is anonymized but could have been submitted as supplementary). The goals of the paper are reasonable and likely to be useful.

### Novelty

The paper addresses important limitations/assumptions of the previous methods and is novel to the best of my knowledge.

### Reproducibility

Hard to evaluate reproducibility without seeing the code and data. The link implies this will be available should the paper be accepted, which is better than not having any code and data.



**Strength And Weaknesses:**

### Strengths

1. The paper aims to address important limitations of past methods (that of forecasting single values rather than distributions, enforcing rigid coherency, and the lack of calibrated forecasts). Addressing these limitations of the previous work are likely to benefit the part of the community interested in these tasks.

2. Empirical evaluation is comprehensive, including several datasets, methods, and metrics.

### Weaknesses

1. Clarity, Formatting Abuse: The paper is hard to read due to various issues in writing, notional imprecision, and formatting abuse (see notes on Clarity below).

2. Reporting the results makes comparisons in a way that the improvement numbers seem large, but the comparisons aren't always reasonable. E.g. an improvement of 550% on log-likelihood is a comparison on an unbounded metric, as compared to, say, accuracy where an improvement in terms of a % improvement carries more meaning. The results would still be meaningful without such % improvement metrics in these cases.

3. It's unclear if a t-test is a reasonable test to compare model performance, specifically that it's unclear why the assumptions required for a t-test hold here.


**Summary Of The Paper:**

This paper aims to address two issues with prior work on the task of hierarchical time series forecasting (HSTF): (i) the assumption of rigid coherency (i.e. forecasts enforce the time-series values of datasets to satisfy the underlying hierarchical constraints strictly) and (ii) the lack of calibrated forecasts.

To address these issues, the paper introduces soft-distributional-coherency regularization (SDCR) that enables end-to-end learning of the forecast distribution at all levels by leveraging information from the underlying hierarchy. The method involves a two-stage process where
a network is trained to output raw predicted distributions for all time series in the dataset. Then a refinement module jointly refines the distribution parameters for all predicted distributions by leveraging the hierarchical relationships between the time series through SDCR.

Evaluation across multiple datasets and missing data indicate that the proposed approach massively outperforms state-of-the-art methods.

**Summary Of The Review:**

The goals of this paper and the proposed method are likely to benefit the community working on hierarchical time series forecasting. The issues involving unclear writing, notational imprecision, and formatting abuse make it hard to read this paper and evaluate the details.

---

### Author Response · Authors · 2022-11-17
**Summary of the changes to the revised submission**

Dear reviewers,

We thank you again for providing valuable comments and recognizing our work’s technical and empirical contributions toward probabilistic hierarchical time-series forecasting.

We wish to summarize the following changes made to our revised submission based on your comments:

1. Clarified the definition of hierarchical relations $H_{\mathcal{T}}$ as $H_{\mathcal{T}} = \{ \mathbf{y}_{i} = \sum_{j \in \mathcal{C}_i} \phi_{ij}\mathbf{y}_{j}: \forall i \in \{1, 2, \dots, N\}, |\mathcal{C}_i|  > 0 \}$ to differentiate between leaf and non-leaf nodes.
2. Revised and clarified some of the notations to implore readability:
   * $D_t$ is replaced by $\mathcal{D}^t$ for consistency
   * Removed bold formatting of subscripts for consistency
   * Clarified the meaning of latent variables in Eq. 2 by adding :
> where $\mathbf{z}_i, \mathbf{u}_i, \mathbf{z}$ are intermediate latent variables of our probabilistic raw forecasting model TSFNP (Section 3.1).
   * Replace $NN$ notation for neural modules with $\Theta$
3. Added that Log-likelihood for LS is capped at -10
4. Fixed *"vspace abuse"* around Figures 1,2, Tables 1,2, and around equations.
5. Replaced \cite with \citep where appropriate
6. Mentioned the potential to extend our work to different kinds of distributions across the hierarchy in the conclusion as suggested by Reviewers UgsV and eMuv
7. Fixed minor typos pointed out by the reviewers

We have also addressed other comments and questions in the replies to the reviews. We hope the reviewers would acknowledge our responses and consider updating their scores. We are happy to answer any further questions or clarifications.

---

> ### Comment · Reviewer_cEHM · 2022-12-05
> **Summary of thoughts on revision**
>
> Hello,
>
> The authors have fixed the notational issues, and several of the fixes seem to be in response to my comments, so I feel I ought to summarize my thoughts on the revision here for the other reviewers and the AC. My response to the authors is here: https://openreview.net/forum?id=YsNlFsG-jj&noteId=AwKMRPGNRw
>
> Pros:
>
> - The idea of this paper is simple and interesting.
> - With fixing the notational issues, the math is explained in a dense manner but at least seems correct based on my re-read.
>
> Cons:
>
> - I retain concerns over the evaluation, especially surrounding the reliability analysis, which cannot be fixed in writing. I've made suggestions for fixing this in my dedicated response.
>
> - I find that the reporting of results and subsequent claims seems to sensationalize large numbers even when reporting them in this manner may not be necessarily meaningful. Some of the claims are not grounded or qualified (e.g. specifying nuances in comparing LS on coherent and incoherent distributions, qualifying how reporting % improvements might not convey to a human reader a clear takeaway when doing so on an unbounded metric like LS).
>
> This contributes to a general trend of lacking clarity in writing. These can be fixed for a camera ready, but from the authors' discussion so far I find it hard to gauge whether they accept this feedback to address it in a camera-ready version.
>
> - The vspace abuse is still glaring and does persist, and I leave this to the AC. [Minor: There are also issues at the bottom of page 7where the ordering in the list of metrics is incorrect (1 > 3 > 2 > 4)].
>
> Overall:
> I am retaining my score of 'barely over acceptance'. The idea might be useful for the community, but the evaluation and quantification may contribute to a noisy trend that other papers carry forward.

---

> ### Comment · Reviewer_UgsV · 2022-12-05
> **Math clarity**
>
> After AC pointed out the ELBO issue to me, I think the clarity in the math is alarming.
>
> In the appendix where the authors derive the ELBO in eq(14), the approximated posterior $q$ is over $z_i$ while in eq(10) the $q$ is over $u_i$. It is also not clear to me how to get eq(10) from eq(15). Further more, in eq(10) there is only $q$ over $u_i$, where is $z_i$?. I have not really worked in the variational inference, maybe I missed something which is obvious to the people in the field. If so, please let me know.
>
> I do think the paper proposed an interesting idea and it makes sense to use soft regularization for the hierarchical constraints than hard ones. The experiments are also comprehensive from my point of view. But these clarity issues in the math, despite the promising empirical results, make me concerned with the soundness of the method. Thus, I don't think I can vote for the paper passing the acceptance bar without these issues being resolved.

---

> > ### Comment · Reviewer_cEHM · 2022-12-06
> > **Filling the gaps and pointing out more minor notational errors in the process**
> >
> > 1\. On the $z_i$ and $u_i$
> >
> > I'm one of the reviewers who complained about clarity, but I think i understand what the authors are trying to say (I did have to do a lot of gap filling myself though.  I assumed the authors mean that they apply a radial basis kernel to find a past $y_j$ similar to a given future $y_i$, and obtain a $\mathbf{z}_i$ through an encoder that accepts as input the average of the most similar past patterns in Eq. 5. Is this correct?
> >
> > Nevertheless, this is really unclearly written. Also, in two lines above Eq. 5, why is the subscript $j$ for $y_j$ roman? What does it mean to sample a $y_j$.
> >
> > With this, the discrepancy between Eq. 10 and Eq. 14 goes away I think.
> >
> > 2\.
> > >  $u$ is the sampled vector embedding of the input time-series from NGP defined in Equation 4
> >
> > The authors are referring to $u_i$, not $u$. The bigger issue is with the notation of $\mu(\mathbf{u})_i$. What does it even mean to put a subscript after the function? The notation implies that $\mu$ is a function that accepts $\mathbf{u}$ as an input. This $\mathbf{u}$ is not well specified.
> >
> > ---
> >
> > I think even with jumping in and making up for unclear writing as a reader, my other larger concerns with evaluation still hold.

---

### Author Response · Authors · 2022-12-08
**Response to recent comments**

We thank the reviewers for engaging with our response. We would like to clarify on the main comments.

## Regarding Math clarity

We thank the reviewer UgsV and AC for reading the derivation in Appendix carefully and for pointing out two typos regarding the ELBO loss. To clarify, $q_i(u_i| y_i^{t’:t})$ term in the variation distribution and Eq. 10 should be replaced by $q_{\phi}(z_i| y_i^{t’:t})$ from Eq. 14. Substituting the variational distribution in Eq 14 into Eq. 15 will give the ELBO loss in Eq 10. We will correct the typos.

Note that these are two symbol typos only. We have gone over the notations multiple times and fixed some issues pointed out by the reviewers. We agree the notations are sometimes dense in the paper but we believe the problem demands it. If there are any further typos, we are happy to fix them. However, we believe that these typos do not invalidate the basic soundness of the entire framework and the math.

## Regarding Eq. 5
We thank the reviewer cEHM for making the effort to understand Eq. 5. The elaboration given by the reviewer regarding Eq. 5 is correct. We did give the intuition in Section 3.1 but skipped the details on the derivation of Eq 5 due to page limit since this is based on prior work [1] (see Page  Section 3.2) and this was not the focus of our work. (Indeed Eq 5 is part of TSFNP model which can be directly substituted by any differentiable probabilistic univariate forecasting model). However, we will be happy to add a few more details on TSFNP in the appendix.

## Regarding notational comments on u
As pointed out in our earlier response $\mu(\mathbf{u})$ and $\sigma(\mathbf{u})$ are not functions of $\mathbf{u}$. They are the parameters of the Gaussian distribution $\mathcal{N}(\mu(\mathbf{u}), \sigma(\mathbf{u}))$ from which we sample $\mathbf{u}$. We believe adding the subscript i as $\mu(\mathbf{u})_i$ vs. $\mu(\mathbf{u}_i)$ comes down to personal taste. We would gladly use the latter or any other alternative if the reviewer thinks it is better. We have italicized all subscripts for consistency and will fix it for y_j as well.

## On “sensational” results
We note that we have used the “550%” number in LS only once in the entire paper (Section 4.2 Q1). We used it simply as done in past works [1,3] (See Section 4.1 in [1] and  Section 5.2 in [3]). We agree with the reviewer that interpreting LS is a bit more nuanced. We will gladly replace it with the absolute difference. Everywhere else, we have used % improvement of CRPS and CS scores only including in the introduction, abstract, and conclusion. We wish to emphasize again that we get substantial improvements across all the benchmarks in our experiments on standard well-known metrics. Hence, based on just this one number in one line, it is a bit unfair to label the entire results as “sensational”.

## On hypothesis testing
We believe the reviewer cHEM conclusion that our results are not ‘reliable’ is unfair.
1. Our scores (in Table 3) clearly show substantial improvement.
2. We followed a reasonable widespread common ML practice for significance testing as also noted by the reviewer.
3. Further, we emphasize that prior works such as MINT, HierE2E, SHARQ and PEMBU do not even perform these significance tests in their paper. Hence we are *already* improving over prior works in the standard for reporting the results.

It is not clear that the reviewer’s proposal is necessarily better. ANOVA can only tell us if at least one of the model’s performances is significantly different compared to others. Indeed, choosing the correct test for different forecast evaluation metrics is a nuanced non-trivial problem (e.g. see [2]) with many proposals (e.g. to also handle sequential correlations between forecasts). Hence, this is not in the scope of our work and we simply used a common practice.

## On vspace issue
Based on reviewers’ feedback we had substantially reduced its usage, specifically around tables, figures and equations in our revised version. We have used \small and \footnotesize on several of the longer equations (Like Eq. 2, 3, 11, 12), following usual past practice. This may make them appear tight around the text. As the reviewers feel this is still distracting, we will gladly submit a version removing all vspace, and full sizing the equations. We hope that these not-hard-to-handle formatting comments do not downweight the research contributions of our work.

[1] Kamarthi et al. Neural non-parametric uncertainty quantification for epidemic forecasting. NeurIPS 2021

[2] Hewamalage, Hansika, Klaus Ackermann, and Christoph Bergmeir. "Forecast Evaluation for Data Scientists: Common Pitfalls and Best Practices." arXiv preprint arXiv:2203.10716 (2022).

[3] Kamarthi et al. "CAMul: Calibrated and Accurate Multi-view Time-Series Forecasting." Proceedings of the ACM Web Conference 2022.

---

### Decision · Program_Chairs · 2023-01-20

**Decision:**

Reject

**Justification For Why Not Higher Score:**

The lack of clarity in the exposition simply make it too hard even for a somewhat informed reader to take much value out of this paper. Clarity is especially important in a field like forecasting, where predictions need to be interpreted, and need to be trustworthy. In fact, it is not that this paper comes in the context of an application where all that matters are good results, and we can just ignore a large set of mostly unmotivated technicalities.


**Justification For Why Not Lower Score:**

N/A


**Metareview: Summary, Strengths And Weaknesses:**

This paper deals with hierarchical forecasting, where observations of real-valued time series are available at different levels of aggregation. The paper stresses the importance of weak consistency, where observed values do not exactly form sums according to the hierarchy, but only approximately so. The authors propose what they call a Bayesian probabilistic model along with a variational inference learning procedure (closely related to prior work), and then extend this optimization problem by constraints coming from the hierarchy. Since these are applied as a form of regularization, the constraints are not enforced as equalities, which allows them to enforce weak consistency. The authors claim that due to their probabilistic model approach, their forecasts are better calibrated than for previous methods.

On the positive side, reviewers state that there could be some useful ideas in this work, which would extend prior work (which is numerous in statistics).

The major downside of this submission is lack of clarity in the presentation, to an extent that makes it hard to judge its value. Just some examples:
- The supposedly probabilistic model is introduced in a long equation with lots of variables that are not defined (u_i, z_i). It is not an equation for a joint probability, because most targets are conditioned on, but some form of predictive. This begs the question how all the posterior terms are being approximated, and what likelihood and model structure linking the latent variables they are based upon
- The authors do not properly explain the relation between u_i and z_i (supposedly both latent variables)
- The variational distributions are not defined, and the paper only talks about distributions over u_i. These equations contain several typos
- The calibration metric used is stated as integral, with undefined functions. It is not clear what is really being evaluated
- There is some talk about sampling something with probability exp(-gamma |u_i - u_j|^2) "into a set N_i", it is totally unclear what is meant here, but this seems an important step when using this model

These are just some examples. One of the reviewers spent several days to try and decode the writing and finally concluded that with a lot of extra, work, the gaps can maybe be filled.

ICLR has a certain bar at clarity and writing quality, and this paper is clearly below this bar. The authors should spend serious efforts on clarifying what they intend to do. Instead of changing the style file to squeeze out every bit of space (as was done in the submission), they should cite earlier closely related work instead of repeating things. If they intend to claim their work represents a proper probabilistic model and a valid variational approximation to it, they should state clearly and precisely what the model is (the joint generative one), and how its latent variables relate to each other, then describe the variational distributions and their structure. They should also state how their constraints or regularization comes in, and what type of dependencies among the variational distributions this implies. Finally, given the high complexity of what is proposed here, compared to previous work, the paper lacks a strong motivation why all this added complexity is needed, in terms of ablations. It also lacks statements on how this complexity is managed, say how the optimization problem is solved reliably, or how long it takes.

Finally, the reviewers also express valid concern about the evaluations.


**Summary Of Ac-Reviewer Meeting:**

- UgsV:
   - Ideas makes sense to me, soft regularization
   - Evaluation comprehensive
   - Not familiar with statistical methods
   - Include MINT, ERM in experiments
   - Model: Architecture, know what they are doing
   - Maths behind it: Makes me worried that something is wrong
- cEHM:
   - Complained about clarity
   - Looked at everything in detail
   - In principle useful
   - Problem in execution
   - Takes a lot of work to even understand, not clarified
   - Sampling with radial basis kernel for similar sequences
   - Evaluation: Does a lot of things (metrics) one should not do
   - Could implement maybe with a lot of pain
   - It is not ready
- B3vd:
   - Surprised we had some much discussion
   - Done quite a lot of work on hierarchical forecasting
   - Reviewed this paper several times
   -May be nice idea, but hard to understand why
   - Could say so many things that are wrong
   - Problem eq. (3): What are the distributions? There is related work
   - Eq. (2): What are all the variables? What does this mean?
- eMuv:
   - Appreciate weak and strong consistency; also end to end
   - Re-read paper: If dataset is weakly consistent, is model as well?
   - Many different metrics: Why this choice?